# Introducing the Video In Situ Snowfall Sensor (VISSS)

Maximilian Maahn[1], Dmitri Moisseev[2,3], Isabelle Steinke[1,*], Nina Maherndl[1], and Matthew D. Shupe[4,5]

[1]Leipzig University, Leipzig Institute of Meteorology (LIM), Germany
[2]Institute for Atmospheric and Earth System Research/Physics, Faculty of Science, University of Helsinki, Finland
[3]Finnish Meteorological Institute, Helsinki, Finland
[4]University of Colorado, Cooperative Institute for Research in Environmental Sciences, Boulder, Colorado, USA
[5]National Oceanographic and Atmospheric Administration, Physical Sciences Laboratory, Boulder, Colorado, USA
[*]Now at Geoscience and Remote Sensing Department, Delft University of Technology, Delft, Netherlands

**Correspondence:** Maximilian Maahn (maximilian.maahn@uni-leipzig.de)

**Abstract.** The open source Video In Situ Snowfall Sensor (VISSS) is introduced as a novel instrument for the characterization of particle shape and size in snowfall. The VISSS consists of two cameras with LED backlights and telecentric lenses that allow accurate sizing and combine a large observation volume with relatively high pixel resolution and a design that limits wind disturbance. VISSS data products include various particle properties such as maximum extent, cross-sectional area, perimeter, complexity, and sedimentation velocity. Initial analysis shows that the VISSS provides robust statistics based on up to 10,000 unique particle observations per minute. Comparison of the VISSS with collocated PIP (Precipitation Imaging Package) and Parsivel instruments at Hyytiälä, Finland, shows excellent agreement with Parsivel, but reveals some differences for the PIP that are likely related to PIP data processing and limitations of the PIP with respect to observing smaller particles. The open source nature of the VISSS hardware plans, data acquisition software, and data processing libraries invites the community to contribute to the development of the instrument, which has many potential applications in atmospheric science and beyond.

## 1 Introduction

It is well known that "every snowflake is unique". The shape of a snow crystal is very sensitive to the processes that were active during its formation and growth. Vapor depositional growth leads to a myriad of crystal shapes depending on temperature, humidity, and their turbulent fluctuations. Aggregation combines individual crystals into complex snowflakes. Riming describes the freezing of small droplets onto ice crystals, causing them to rapidly gain mass and form a more rounded shape. In other words, the shape of snow particles is a fingerprint of the dominant processes during the life-cycle of snowfall.

Better observations of the fingerprints of snowfall formation processes are needed to advance our understanding of ice and mixed-phase clouds and precipitation formation processes (Morrison et al., 2020). Given the importance of snowfall formation processes for global precipitation (Mülmenstädt et al., 2015; Field and Heymsfield, 2015), the lack of process understanding leads to gaps in the representation of these processes in numerical models. In a warming climate, precipitation amounts and extreme events, including heavy snowfall, are expected to increase (Quante et al., 2021), but the exact magnitudes are associated with large uncertainties (Lopez-Cantu et al., 2020).

Remote sensing observations of snowfall are indirect, which limits their ability to identify snow particle shape by design. Ground-based in situ observations of ice and snow particles can identify the fingerprints of the snowfall formation processes and provide detailed information on particle size, shape, and sedimentation velocity. Using assumptions about sedimentation velocity or an aggregation and riming model as a reference, the particle mass-size and/or density relationship can also be inferred from in situ observations. (Tiira et al., 2016; von Lerber et al., 2017; Pettersen et al., 2020; Tokay et al., 2021; Leinonen et al., 2021; Vázquez-Martín et al., 2021a). Various attempts have been made to classify particle types and identify active snowfall formation processes using various machine learning techniques (Nurzyńska et al., 2013; Grazioli et al., 2014; Praz et al., 2017; Hicks and Notaroš, 2019; Leinonen and Berne, 2020; Del Guasta, 2022; Maherndl et al., 2023b); these classifications are needed to support quantification of snowfall formation processes (Grazioli et al., 2017; Moisseev et al., 2017; Dunnavan et al., 2019; Pasquier et al., 2023). In situ observations have also been used to characterize particle size distributions (Kulie et al., 2021; Fitch and Garrett, 2022), investigate sedimentation velocity and turbulence of hydrometeors (Garrett et al., 2012; Garrett and Yuter, 2014; Li et al., 2021; Vázquez-Martín et al., 2021b; Takami et al., 2022), and for model evaluation (Vignon et al., 2019). In combination with ground-based remote sensing, in situ snowfall data have been used to validate or better understand remote sensing observations (Gergely and Garrett, 2016; Li et al., 2018; Matrosov et al., 2020; Luke et al., 2021), to develop joint radar in situ retrievals (Cooper et al., 2017, 2022), and to train remote sensing retrievals (Huang et al., 2015; Vogl et al., 2022).

Different design concepts have been used for in situ snowfall instruments. Line scan cameras are commonly used by optical disdrometers such as the OTT Parsivel (Löffler-Mang and Joss, 2000) and their relatively large observation volume reduces the statistical uncertainty for estimating the particle size distribution (PSD). However, additional assumptions are required to size irregularly shaped particles such as snow particles correctly due to the one-dimensional measurement concept (Battaglia et al., 2010). This limitation can be overcome when adding a second line camera as for the 2DVD (2-dimensional video disdrometer, Schönhuber et al., 2007), but particle shape estimates can still be biased by horizontal winds (Huang et al., 2015; Helms et al., 2022). The 2DVD's pixel resolution of approx. 190 μm per pixel (px) and the lack of gray-scale information prohibits resolving fine-scale details of snow particles.

To get high resolution images, a group of instruments uses various approaches to obtain particle images with microscopic resolution at the expense of the measurement volume size. For example, the MASC (Multi-Angle Snowfall Camera, Garrett et al., 2012) takes three images with 30 μm px$^{-1}$ pixel resolution of the same particle from different angles. This allows for resolving very fine particle structures, but during a snowfall event Gergely and Garrett (2016) observed only $10^2$ - $10^4$ particles which is not sufficient to reliably estimate a PSD on minute temporal scales needed to capture changes in precipitation properties. Del Guasta (2022) have developed a flatbed scanner (ICE-CAMERA) that has a pixel resolution of 7 μm px$^{-1}$ and can provide mass estimates by melting the particles, but this approach only works at low snowfall rates. The images of the D-ICI (Dual Ice Crystal Imager, Kuhn and Vázquez-Martín, 2020) have even a pixel resolution of 4 μm px$^{-1}$ and show particles from two perspectives, but similar to the MASC, the small sampling volume does not allow for the measurement of PSDs with a sufficiently high accuracy.

The SVI (Snowfall Video Imager, Newman et al., 2009) and its successor the PIP (Precipitation Imaging Package, Pettersen et al., 2020) use a camera pointed to a light source to image snow particles in free fall. The open design limits wind field perturbations and the large measurement volume (4.8 x 6.4 x 5.5 cm for a 1 mm snow particle) minimizes statistical errors in deriving the PSD. However, the pixel resolution of 100 μm px$^{-1}$ is not sufficient to study fine details. Further, the open design requires that the depth of the observation volume is not constrained by the instrument itself. As a consequence, particle blur needs to be used to determine whether a particle is in the observation volume or not which is potentially more error prone than a closed instrument design. A similar design was used by Testik and Rahman (2016) to study the sphericity oscillations of raindrops. Kennedy et al. (2022) developed the low-cost OSCRE (Open Snowflake Camera for Research and Education) system that uses a strobe light to illuminate particles from the side allowing for the observation of particle type of blowing and precipitating snow but the observation volume is not fully constrained.

This study presents the Video In Situ Snowfall Sensor (VISSS). The goal was to develop a sensor with an open instrument design without sacrificing the quality of measurement volume definition or resolution. It uses the same general principle as the PIP (Fig. 1): gray-scale images of particles in free fall illuminated by a background light. Unlike the PIP, this setup is duplicated with overlapping measurement volumes so that particles are observed simultaneously from two perspectives at a 90° angle. This robustly constrains the observation volume without the need for further assumptions. In addition, having two perspectives of the same particle increases the likelihood that the observed maximum dimension ($D_{max}$) and aspect ratio are representative of the particle. While the VISSS does not reach the microscopic resolution of the D-ICI or ICE-CAMERA, its pixel resolution of 43 to 59 μm px$^{-1}$ is significantly better than the PIP, and the use of telecentric lenses eliminates sizing errors caused by the variable distance of snow particles to the cameras.

The VISSS was originally developed for the MOSAiC (Multidisciplinary drifting Observatory for the Study of Arctic Climate) experiment (Shupe et al., 2022) and deployed at MetCity and, after the sea ice became too unstable in April 2020, on the P-deck of the research vessel Polarstern. After MOSAiC, the original VISSS was deployed at Hyytiälä, Finland (Petäjä et al., 2016) in 2021/22 and at Gothic, Colorado as part of the SAIL campaign in 2022/23 (Surface Atmosphere Integrated Field Laboratory, Feldman et al., 2021). During a test setup in Leipzig, Germany, the VISSS was used to evaluate a radar-based riming retrieval (Vogl et al., 2022). An improved second generation of VISSS was installed at the French-German Arctic research base AWIPEV (the Alfred Wegener Institute Helmholtz Centre for Polar and Marine Research - AWI - and the French Polar Institute Paul Emile Victor - PEV) in Ny-Ålesund, Svalbard (Nomokonova et al., 2019) in 2021. A further improved third generation VISSS is currently being built at the Leipzig University and will be deployed in Hyytiälä end of 2023. The VISSS hardware plans and software libraries have been released under an open source license (Maahn et al., 2023; Maahn, 2023a, b) so that the community can replicate and further develop VISSS. The VISSS hardware design and data processing are described in Sects. 2 and 3, respectively. Example cases including a comparison with the PIP are given in Sect. 4 and concluding remarks are given in Sect. 5.

## 2 Instrument design

The VISSS consists of two camera systems oriented at a 90° angle to the same measurement volume (Fig. 1). Both cameras work using the Complementary Metal Oxide Semiconductor (CMOS) global shutter principle and use a resolution of 1280x1024 gray-scale pixels and a frame rate of 140 Hz (250 Hz since the 2nd generation). One camera acts as the leader, sending trigger signals to both the follower camera and the two LED backlights that illuminate the scenes from behind with a 350,000 lux flash. Green backlights (530 nm) were chosen because the camera and lenses are optimized for visual light. The leader-follower setup results in a slight delay in the start of exposure between the two cameras. To compensate for this, the background LEDs are turned on for a duration of 60 μs only when the exposure of both cameras is active. Thus, the 60 μs flash of the backlights determines the effective exposure time of the camera as long as there is no bright sunlight, which is a rare condition during precipitation.

The two camera-lens-backlight combinations are at a 90° angle so that particles are observed from two perspectives, reducing sizing errors. Leinonen et al. (2021) found that using only a single perspective for sizing snow particles can lead to a normalized root mean square error of 6% for $D_{max}$ and Wood et al. (2013) estimated the resulting bias in simulated radar reflectivity to be 3.2 dB. For the VISSS, the accuracy of the measurements can be further improved by taking advantage of the fact that the VISSS typically observes 8 to 11 frames of each particle (assuming a sedimentation velocity of 1 m s$^{-1}$ and a frame rate of 140 to 250 Hz), and additional perspectives can be obtained from the natural tumbling of the particle.

Telecentric lenses have a constant magnification within the usable depth of field, eliminating sizing errors. Consequently, the lens aperture must be as large as the observation area, making the lens bulky, heavy and expensive. For the first VISSS (VISSS1), a lens with a magnification of 0.08 was chosen, resulting in a pixel resolution of 58.75 μm px$^{-1}$ (Table 1). The working distance, i.e. the distance from the edge of the lens to the center of the observation volume, is 227 mm. This partly undermines the goal of having an instrument with an observation volume that is not obstructed by turbulence induced by nearby structures, but was caused by budget limitations. It also does not allow for sufficiently large roofs over the camera windows to protect against snow accumulation in all weather conditions. This problem was partially solved by the increased budget (22 kEUR) for the second generation VISSS2, which used a 600 mm working distance lens as well as a camera with an increased frame rate of 250 Hz and a pixel resolution of 43.125 μm px$^{-1}$. However, the optical quality of the lens proved to be borderline for the applications, resulting in an estimated optical resolution of approximately 50 μm and slightly blurred particle images. Consequently, the lens was changed again for the third generation VISSS3 (currently under construction), which has a working distance of 1300 mm. This was motivated by the result of Newman et al. (2009) that the air flow is undisturbed at a distance of 1 m from the instrument. Image quality is potentially also impacted by motion blur and the exposure time of 60 μs was selected to limit motion blur of particles falling at 1 m/s to 1.02 and 1.44 pixels for VISSS1 and VISSS2, respectively. Particle blur can also occur when particles are not exactly in focus of the lenses. The maximum circle of confusion is 1.3 pixels at the edges of the observation volume.

The lens-camera combinations and backlights are housed in waterproof enclosures that are heated to −5°C and 10°C, respectively. The low temperature in the camera housing is to prevent melting and refreezing of particles on the camera window.

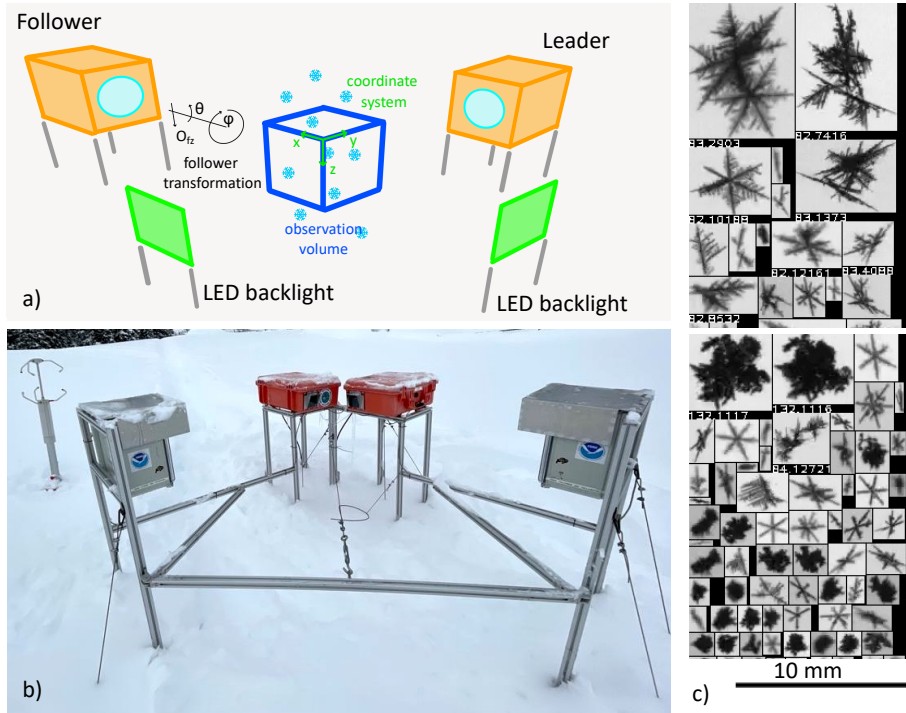

**Figure 1.** a) Concept drawing of the VISSS (not to scale with enlarged observation volume). See Sections 3.2 and 3.3 for a discussion of the joint coordinate system and the transformation of the follower's coordinate system, respectively. b) First generation VISSS deployed at Gothic, Colorado during the SAIL campaign (Photo by Benn Schmatz), c) Randomly selected particles observed during MOSAiC on 15 November 2019 between 6:53 and 11:13 UTC.

The cameras of VISSS1 and VISSS2 are connected to the data acquisition systems via separate 1 Gbit and 5 Gbit Ethernet connections, respectively. Due to the increased frame rate, two separate systems are required to record data in real-time for VISSS2.

## 3   Data processing

The cameras transmit every captured image to the data acquisition systems which are standard desktop computers running Linux. Based on simple brightness changes, the computers save only moving images and discard all other data (this was not implemented for MOSAiC yet). The raw data of the VISSS consists of the video files (mov or mkv video files with h264 compression), the first recorded frame as an image (jpg format) for quick evaluation of camera blocking, and a csv file with the timestamps of the camera (capture_time) as well as the computer (record_time) and other meta information for each frame. The cameras run continuously and new files are created every 10 minutes (5 minutes for MOSAiC). In addition, a daily status csv file is maintained that contains information about software start and stop times and when new files were created. Both cameras

**Table 1.** Technical specifications of the three VISSS instruments.

| | VISSS1 | VISSS2 | VISSS3 |
|---|---|---|---|
| Pixel resolution [µm px$^{-1}$] | 58.75 | 43.125 | 46.0 |
| Obs. volume (w x d x h) [mm] | 75.2 x 60.1 x 60.1 | 55.2 x 44.2 x 44.2 | 58.9 x 47.1 x 47.1 |
| Used frame size [px] | 1280 x 1024 | 1280 x 1024 | 1280 x 1024 |
| Frame rate [Hz] | 140 | 250 | 250 |
| Effective exposure time [µs] | 60 | 60 | 60 |
| Working distance [mm] | 227 mm | 600 mm | 1300 mm |
| Camera | Teledyne Genie Nano M1280 Mono | Teledyne Genie Nano 5G M2050 Mono | Teledyne Genie Nano 5G M2050 Mono |
| Lens | Opto Engineering TC12080 | Sill S5LPJ1235 (with modified working distance) | Sill S5LPJ1725 (with modified working distance) |
| Maker | University of Colorado Boulder | University of Cologne | Leipzig University |
| Deployments | MOSAiC 2019/20, Hyytiälä 2021/22, SAIL 2022/23 | Ny-Ålesund since 2021 | Hyytiälä (planned for winter 2023/24) |

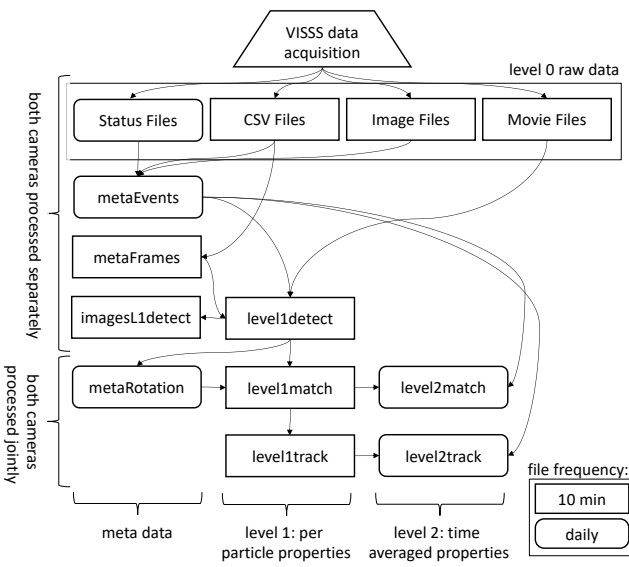

**Figure 2.** Flowchart of VISSS data processing. Daily products have rounded corners, 10-minute resolution products have square corners.

record completely separately which requires an accurate synchronization of the camera and computer clocks for matching the observations of a single particle.

Obtaining particle properties from the individual VISSS video images requires (1) detecting the particles, (2) matching the observations of the two cameras, and (3) tracking the particles over multiple frames to estimate the fall velocities. The level 1

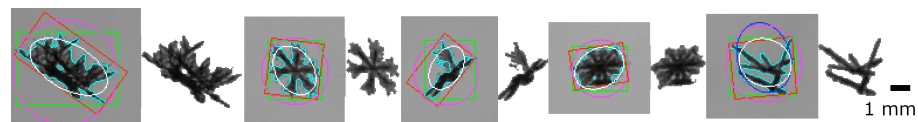

**Figure 3.** Estimation of particle perimeter $p$ and area $A$ (cyan), maximum dimension $D_{max}$ (via smallest enclosing circle, magenta), smallest rectangle (red), region of interest ROI (green), and elliptical fits using openCV's fitEllipseDirect (white) and fitEllipse functions (blue, covered by white line if identical to fitEllipseDirect). The particles were observed during MOSAiC on 15 November 2019 05:25 UTC except the particle on the right (Hyytiälä 23 January 2022 04:10 UTC).

products contain per-particle properties in pixel units using (1) a single camera, (2) matched particles from both cameras, and (3) exploiting particles tracked in time. For the level 2 products, the level 1 observations are calibrated (i.e., converted from pixel in metric units) and distributions of particle size, aspect ratio, and other properties are estimated based on the per-particle properties. In addition to the level 1 and level 2 products, there are metadata products: metaEvents is a netcdf version of the status files along with a camera blocking estimate based on the jpg images. metaFrames is a netcdf version of the csv file. metaRotation keeps track of the camera misalignment as detailed below. The imagesL1detect product contains images of the detected particles which is required for creating quicklooks like Fig. 1.c.

In the following, the processing of the level 1 and level 2 products is described in detail (Fig. 2).

## 3.1 Particle Detection

Hydrometeors need to be detected and sized based on individual frames. First, video frames containing motion are identified by a simple threshold-based filter. Except for the MOSAiC dataset, this is done in real-time, which significantly reduces the data volume. Because snow may stick to the camera window, individual particles within a video frame cannot be identified by image brightness. Instead, the moving region of interest (ROI) is identified by openCV's BackgroundSubtractorKNN class (Zivkovic and van der Heijden, 2006) in the image coordinate system (horizontal dimension $X$, vertical dimension $Y$ pointing to the ground). The moving mask identified by the background subtraction methods cannot be used directly for particle detection because the particles in the moving foreground mask are systematically too large. For each particle, we select a 10 pixel padded box around the region of interest (ROI) which is the smallest non-rotated rectangular box around the particle (Fig. 3). Then, we use openCV's Canny edge detection (after applying a Gaussian blur with a standard deviation of 1.5 pixels) to identify the edges of the particle and the corresponding particle masks. To fill in small gaps in the particle contour, we dilate the contour by 1 pixel, fill the contour, erode by 1 pixel, and identify the new contour. This method closes potential holes in the particle mask that should be retained to avoid overestimation of particle area. Therefore, the final particle mask contains only values confirmed by the Canny filter and the background detection mask. As a result, VISSS can detect even relatively small particle structures, as shown in Fig. 3. The use of only 1 pixel (i.e., 43 to 59 μm) for dilation was found to be sufficient and allows to potentially resolve more details of the particles than MASC and PIP, which dilate by 200 μm (Garrett et al., 2012) and 300 μm (Helms et al., 2022), respectively. The final particle mask and contour are used to estimate the particle's maximum dimension

(using openCV's minEnclosingCircle function), perimeter $p$ (arcLength), area $A$ (contourArea) and aspect ratio $AR$ (defined as the ratio between the major and minor axis), as well as the canting angle $\alpha$ (defined between vertical axis and major axis).

$AR$ and $\alpha$ are estimated in three different ways, from the smallest rectangle fitted around the contour (minAreaRect) or from an ellipse fitted to the contour (fitEllipse and the more stable fitEllipseDirect). Particle area equivalent diameter ($D_{eq}$) is obtained from $A$. Particle complexity $c$ (Garrett et al., 2012; Gergely et al., 2017) is derived from the ratio between particle perimeter $p$ to the perimeter of a circle with same area $A$

$$c = \frac{p}{2\sqrt{\pi A}}. \tag{1}$$

In addition to these geometric variables, the level1detect product contains variables describing the pixel brightness (min, max, standard deviation, mean, skewness), the position of the centroid, and the blur of the particle estimated from the variance of the Laplacian of the ROI. All particles are processed for which $D_{max} \geq 2$ px and $A \geq 2$ px holds. To avoid detection of particles completely out of focus, the brightness of the darkest pixel must be at least 20 steps darker than the median of the entire image and the variance of the Laplacian of the ROI brightness must be at least 10. Particle detection is the most computationally intensive processing step and is typically performed on a small cluster. Processing 10 minutes of heavy snowfall for a single VISSS camera can take several hours on a single AMD EPYC 7302 core.

### 3.2 Particle Matching

The particle detection of each camera is completely separate, so the particles observed by each camera must be combined. This particle combination allows for the particle position to be determined in a three-dimensional reference coordinate system. As a side effect, this constrains the observation volume by discarding particles outside of the intersection of their observation volumes, i.e. observed by only one camera. We use a right-handed reference coordinate system ($x$,$y$,$z$) with $z$ pointing to the ground to define the position of particles in the observation volume (Fig. 1). In the absence of an absolute reference, we attach the coordinate system to the leader camera (i.e., ($x_L$,$y_L$,$z_L$) = ($x$,$y$,$z$)) such that $x = X_L$ and $z = Y_L$, where $X_L$ and $Y_L$ are the particle positions in the two dimensional leader images. Note that small letters describe the three dimensional coordinate system and capital letters describe the two dimensional position on the images of the individual camera images. The missing dimension $y$ is obtained from the follower camera with $y = -X_F$ where $X_F$ the horizontal position in the follower image.

The matching of the particles from both cameras is based on the comparison of two variables: The vertical position of the particles and their vertical extent. Due to measurement uncertainties, the agreement of these variables cannot be perfect and they are treated probabilistically. That is, it is assumed that the difference in vertical extent $\Delta h$ (vertical position $\Delta z$) between the two cameras follows a normally distributed probability density function (PDF) with mean zero and standard deviation 1.7 px (1.2 px), based on an analysis of manually matched particle pairs. Since pixel measurements are discrete with 1 px steps, the PDF is integrated for an interval of $\pm$ 0.5 px.

This process requires matching the observations of both cameras in time. The internal clocks of the cameras ("capture time") can deviate by more than 1 frame per 10 minutes. The time assigned by the computers ("record time") is sometimes, but not

always, distorted by computer load. Therefore, the continuous frame index ("capture id") is used for matching, but this requires determining the index offset between both cameras at the start of each measurement (typically 10 minutes). For this, the algorithm uses pairs of frames with observed particles that are less than 1 ms (i.e. less than 1/4 of the measurement resolution) apart in record time assuming that the lag due to computer load is only sporadically increased. This allow to identify the most common capture id offset of the frame pairs. We found that this method gives already stable results for a subset of 500 frames.

Similar to $h$ and $z$, the capture id offset $\Delta i$ is used as the mean of a normal distribution with a standard deviation value of 0.01, which ensures that only particles observed at the same time are matched. During MOSAiC, the data acquisition computer CPUs turned out to be too slow to keep up with processing during heavy snowfall. With the additional impact of a bug in the data acquisition code and drifting computer clocks when the network connection to the ship's reference clock were interrupted, the particle matching for the MOSAiC data set often requires manual adjustment. These problems have been resolved for later campaign so that matching now works fully automatic.

The joint product of the probabilities from $\Delta h$, $\Delta z$, and $\Delta i$ is considered a match score, which describes the quality of the particle match. Manual inspection revealed that the number of false matches increases strongly for match scores less than 0.001, which is used as a cut-off criterion. Assuming that the probabilities are correctly determined, this implies that 0.1% of particle matches are falsely rejected, resulting in a negligible bias.

For each particle, its three-dimensional position is provided and all per-particle variables from the detection are carried forward to the matched particle product level1match. The ratio of matched to observed particles from a single camera varies with the average particle size, since larger particles can be identified even when they are out of focus, and varies between approximately 10% and 90%.

## 3.3 Correction for camera alignment

Although alignment of both observation volumes is a priority during installation, the cameras can be rotated or displaced, i.e., misaligned. As a result, the same particle may be observed at different heights and $z = Y_L = Y_F$ does not hold. The observed offsets are not constant and can change due to unstable surfaces or pressure of accumulated snow on the VISSS frame. We could simply ignore the misalignment and continue to take $z$ from the leader, but this would not allow to generally use the vertical position to match particles from both cameras (see above). Also, offsets in $z$ reduce the common observation volume of both cameras, which could lead to biases when calibrating the PSDs if not accounted for.

Besides a constant offset in the vertical $z$ dimension $O_{fz}$, one of the cameras can also be rotated around the optical axis (expressed analogously to aircraft coordinate systems with roll $\varphi$), around the horizontal axis perpendicular to the optical axis (pitch $\theta$), or around the vertical axis (yaw $\psi$). As a consequence, $\Delta z = Y_L - Y_F$ depends on the position of the particle in the observation volume.

To account for the misalignment, we attach the coordinate system to the leader (i.e., we assume that the leader is perfectly aligned $(x_L, y_L, z_L) = (x, y, z)$) and retrieve the misalignment of the follower with respect to the leader in terms of $\varphi$, $\theta$ and $O_{fz}$. We cannot derive $\psi$ from the observation and we have no choice but to neglect it by assuming $\psi = 0$ to reduce the number of unknowns. Mathematically, we need to transform the follower coordinate system $(x_F, y_F, z_F)$ to our leader reference coordinate

system ($x_L$,$y_L$,$z_L$) using rotation and shear matrices. In the appendix A, we show how the transformation matrices can be arranged so that the follower's vertical measure $z_F$ can be converted to $z_L$ depending on $\varphi$ and $\theta$ with

$$z_L = -\frac{\sin\theta}{\cos\theta}x_L + \frac{\sin\varphi}{\cos\theta}y_F + \frac{\cos\varphi}{\cos\theta}(z_F + O_{fz}). \tag{2}$$

This equation can be considered as a forward operator that calculates the expected leader observation $z_L$ based on a misalignment state ($O_{fz}$, $\varphi$, and $\theta$) and additional parameters ($x_L$, $y_F$, $z_F$). While we assume that the misalignment state is constant for each 10 minute observation period, the other variables ($x_L$, $y_F$, $z_F$) are available on a per-particle basis, combining observations from both cameras. Therefore, we can use a Bayesian inverse Optimal Estimation retrieval (Rodgers, 2000) implemented by the pyOptimalEstimation library (Maahn et al., 2020) to retrieve the misalignment state from the actual observed $z_L$.

The retrieved misalignment parameters are required for matching, but retrieving the misalignment parameters requires matched particles. To solve this dilemma, we use an iterative method assuming that misalignment does not change suddenly. The method starts by using the misalignment estimates and uncertainties (inflated by a factor of 10) from the previous time period (10 minutes) to match particles of the current time period. These particles are used to retrieve values for $\varphi$, $\theta$, and $O_{fz}$ which are used as a priori input for the next iteration of misalignment retrieval. The iteration is stopped when the changes in $\varphi$, $\theta$, and $O_{fz}$ are less than the estimated uncertainties. For efficiency, the iterative method is applied only to the first 300 observed particles and the resulting coefficients are stored in the metaRotation product. A drawback of the method is that this processing step requires processing the 10-minute measurement chunks in chronological order, creating a serial bottleneck in the otherwise parallel VISSS processing chain. Obviously, this method does not work when no information is available from the previous time step, e.g., after the instrument was set up or adjusted. To get the starting point for the iteration, the matching algorithm is applied for frames where only a single, relatively large ($> 10$ px) particle is detected, so that the matching can be done based on particle height difference ($\Delta h$) alone, ignoring vertical offset ($\Delta z$).

### 3.4 Particle Tracking

Tracking a matched particle over time provides its three-dimensional trajectory, from which sedimentation velocity and interaction with turbulence can be determined. Since the natural tumbling of the particles provides new particle perspectives, the estimates of particle properties such as $D_{max}$, $A$, $p$, and $AR$ can be further improved. This can be seen in a composite of a particle (Fig. 4.a-b) observed during MOSAiC, which also shows how the multiple perspectives of the particle help to identify its true shape. The example also shows that during MOSAiC the alignment of the cameras was not perfect, resulting in some of the measurements being slightly out of focus; this has been resolved for later campaigns. The tracking algorithm uses a probabilistic approach similar to particle matching taking into account that the particles' velocities only change to a certain extent from one frame to the next. That change can be quantified as a cost derived from the particles' distances and shape differences between two time steps. This allows to use the Hungarian method (Kuhn, 1955) to assign the individual matched particles to particle tracks for each time step in a way that minimizes the costs, i.e. to solve the assignment problem. To account for the fact that the particle's position is expected to change between observations, we use a Kalman filter (Kalman, 1960) to predict a particle's position based on the past trajectory and use the distance $\delta l$ between predicted and actual position for the

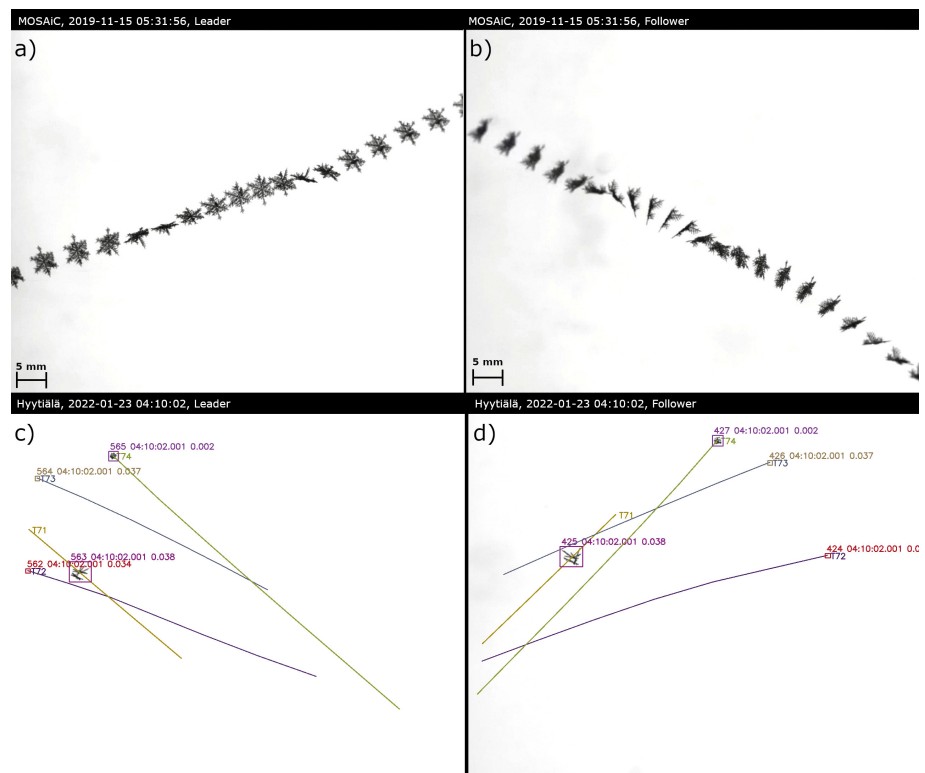

**Figure 4.** Composit of a snow particle recorded by leader (a) and follower (b) during MOSAiC on 15 November 2019 05:31 UTC. Result of particle tracking for the leader (c) and follower (d) of a snow particles recorded in Hyytiälä on 23 January 2022 04:10 UTC. The tracks indicate past and future positions of a particle and are labeled. with the track id number starting with T. Only parts of the tracks observed by both cameras are displayed. The number at the particles denote particle id, time of observation, and match score.

cost estimate. Without a past trajectory, the Kalman filter uses a first guess which we derive from the velocities of previously tracked particles. We found that tracking based only on position is unstable and added the difference of particle area ($\delta A$, mean of both cameras) to the cost estimate to promote continuity of particle shape. The combined cost is estimated from the product

of $\delta l$ and $\delta A$ weighted by their expected variance. The performance of the algorithm can be seen for an observation obtained in Hyytiälä on 23 January 2022 04:10 UTC where multiple particles are tracked at the same time (Fig. 4.c-d). The results of the tracking algorithm are stored in the level1track product which contains the track id and the same per particle variables as the other level 1 products.

### 3.5 Particle size distributions

To estimate the particle size distribution (PSD), i.e., the particle number concentration as a function of size, the individual particle data are binned by particle size (1 px spacing, i.e. 43.125 or 58.75 μm) and averaged to one minute resolution for particle properties such as size, area, and perimeter. These level 2 products are available based on the level1match and level1track

products. For level2match, binned particle properties are available either from one of the cameras or using the minimum, average or maximum from both cameras for each observed particle property. This means that the multiple observations of the same particle all contribute to the PSD. This does not bias the PSD because the number of observed particles is divided by the number of frames, and the PSD describes how many particles are *on average* in the observation volume. For level2track, the distributions are based on the observed tracks instead of individual particles, and are calculated using the minimum, maximum, mean, or standard deviation along the observed track using both cameras. The use of the maximum (minimum) value along a track is motivated by the assumption that the estimated properties of a particle such as $D_{max}$ ($AR$) of a particle will be closer to the true value than when ignoring the different perspectives of a particle along the track obtained by the two cameras.

For both level2 variants, the binned PSD and $A$, perimeter $p$, and particle complexity $c$ are available binned with $D_{max}$ and $D_{eq}$ to allow comparison with instruments using either size definition. In addition to the distributions, PSD-weighted mean values are available for $A$, $AR$, and $c$ in addition to the first to fourth and sixth moments of the PSD that can be used to describe normalized size distributions (Delanoë et al., 2005; Maahn et al., 2015).

For VISSS observations where only a single camera is available, it would also be possible to develop a product based on particles detected by a single camera, using a threshold based on particle blur to define the observation volume, similar to the PIP (Newman et al., 2009).

### 3.6 Calibration

Calibration is required to convert $D_{max}$, $D_{eq}$, and $p$ from pixels to µm. It depends not only on the optical properties of the lens but also on the used computer vision routines. Calibration is obtained using reference steel or ceramic spheres with 1 to 3 mm diameter that are dropped into the VISSS observation volume. After processing using the standard VISSS routines, the estimated sizes are compared to the expected ones. A linear least square fit is applied to the 604 reference sphere observations obtained at Hyytiälä and SAIL resulting in

$$D_{max}[px] = (0.01700 \pm 0.00001) \cdot D_{max}[\mu m] + (0.49301 \pm 0.02101), \tag{3}$$

for the VISSS1 (Fig. 5.a) and

$$D_{max}[px] = (0.02311 \pm 0.00003) \cdot D_{max}[\mu m] + (0.81569 \pm 0.06997), \tag{4}$$

for the VISSS2 based on 372 samples from Ny-Ålesund (Fig. 5.b). The inverse of the slope is 58.832 µm px$^{-1}$ (43.266 µm px$^{-1}$) and is close to the manufacturer's specification of 58.75 µm px$^{-1}$ (43.125 µm px$^{-1}$) for the VISSS1 (VISSS2). The random error estimated from the normalized root mean square error obtained from the difference between observed and expected size is less than 0.8% indicating that random errors are negligible. To investigate the source of the non-zero intercept, we also tested the VISSS computer vision routines with artificially created VISSS images with drawn spheres and compared the expected to measured $D_{max}$ by a least squares fit (Fig. 5.c). Gaussian blur with a standard deviation between 0 and 3 px

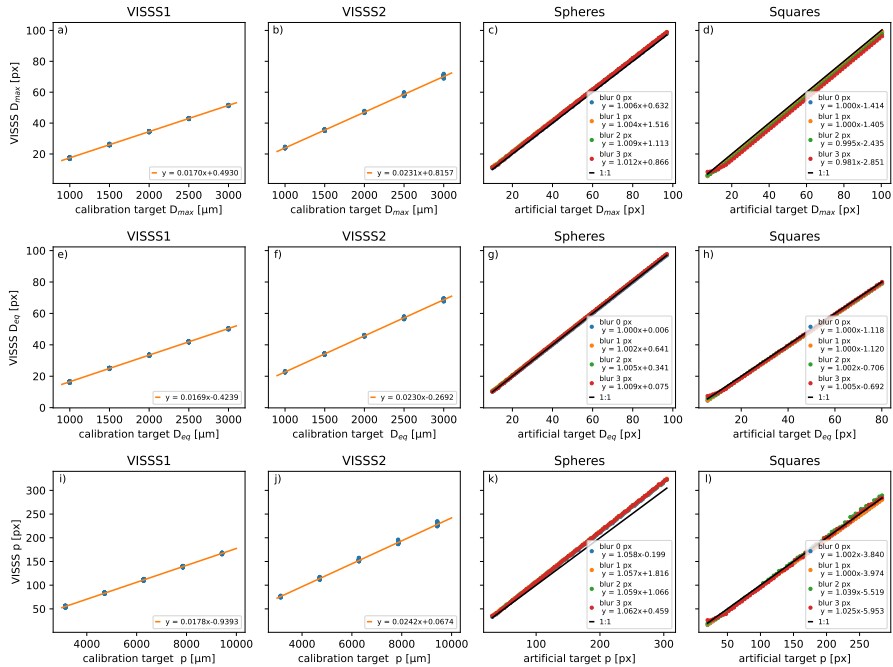

**Figure 5.** Calibration of $D_{max}$ (first row), $D_{eq}$ (second row), and perimeter $p$ (third row) using metal spheres for the VISSS 1 (first column), using metal spheres for the VISSS 2 (second column), using artificial sphere images (third column), and using artificial square images (fourth column). For artificial images, a Gaussian blur filter is applied with a standard deviation according to the embedded legends. The legends also show the results of linear least squares fits.

was applied to account for a realistic range of blurring due to e.g., motion blur or particles that are slightly out of focus. Note that in addition to that, a Gaussian blur filter with a standard deviation of 1.5 px needs to be applied during image processing for the Canny edge detection as discussed above. For the artificial spheres, the obtained slope deviates less than 2% from the expected slope of 1.0, but the offset ranges from 0.6 to 1.5 px caused by the seeming enlargement of the particle due to the applied blur. To investigate the shape dependency of the results, we repeated the experiment with squares (Fig. 5.d). Again, the slope deviates less than 2% from 1.0, but the offset is this time negative with values ranging between $-1.4$ px and $-2.9$ px depending on blur. This is because the corners of the square are rounded when applying Gaussian blur so that the true $D_{max}$ can no longer be obtained. In summary, the VISSS routines overestimate $D_{max}$ of spheres, but underestimate $D_{max}$ of squares. In reality, the VISSS observes a wide range of different shapes that can be both rather spherical or rather complex with "pointy" corners. Therefore, we decided to set the intercept to 0 when calibrating $D_{max}$ which can cause a particle shape dependent bias of $\pm 4$ to $\pm 6\%$. For particles smaller than 10 px, this bias can be slightly larger due to discretization errors as can be seen from the larger impact of blur for small squares (Fig. 5.d).

For better comparison with $D_{max}$, $D_{eq}$ is used instead of $A$ for testing the computer vision method for estimating $A$ (Fig 5.e-h). The results are almost identical to $D_{max}$ so that the slopes derived from $D_{max}$ are applied to $D_{eq}$ (and consequently $A$) as well.

For the perimeter $p$ (Fig. 5.i-l), the slopes derived from the reference spheres are about 5% steeper than for $D_{max}$ indicating that VISSS $p$ are biased high. This bias is also found for artificial spheres independent of the applied additional blur. Therefore, this bias is related to the image processing and most likely caused by the Gaussian blur required for the Canny edge detection. For squares, however, the slope is close to 1 likely due to compensating effects caused by "cutting corners" of the algorithm. In reality, the VISSS observes more complex particles for which the perimeter increases with decreasing scale. (compare to coast line paradox, Mandelbrot, 1967). Therefore, we conclude that it is extremely unlikely that the perimeter of real particles is biased high like for artificial spheres but rather biased low depending on complexity. As a pragmatic approach, we also apply the $D_{max}$ slope to $p$ but stress that $p$ has a considerably higher uncertainty than $D_{max}$ or $D_{eq}$.

The calibration is also checked by holding a millimeter pattern in the camera and measuring the pixel distance in the images, the found difference to the reference spheres is less than 2%. The millimeter pattern calibration did not reveal any dependence on the position in the observation volume so that errors related to imperfect telecentricity of the lenses can be likely neglected.

Calibration of the PSD also requires to obtain the exact size of the observation volume. For perfectly aligned cameras, this would simply be the volume of a rectangular cuboid with a base of 1280 px x 1280 px and a height of 1024 px. However, due to misalignment of the cameras, the actual joint observation volume is slightly smaller than a rectangular cuboid and can have an irregular shape. Therefore, the observation volumes are first calculated separately for leader and follower. To calculate the intersection of the two individual observation volumes, the eight vertices of the follower observation volume are rotated to the leader coordinate system, and the OpenSCAD library is used to calculate the intersection of the two separate observation volumes. To account for the removal of partially observed particles detected at the edge of the image, the effective observation volume is reduced by $D_{max}/2$ px on all sides. Finally, the volume is converted from pixel units to m$^3$ using the calibration factor estimated above.

## 4 Pilot studies

Here, we analyze first generation VISSS (VISSS1) data collected in winter 2021/22 at the Hyytiälä Forestry Field Station (61.845°N, 24.287°E, 150 m MSL) operated by the University of Helsinki, Finland to show the potential of the instrument. For comparison, we use a co-located PIP (von Lerber et al., 2017; Pettersen et al., 2020) and OTT Parsivel[2] laser disdrometer (Löffler-Mang and Joss, 2000; Tokay et al., 2014). The distance between the VISSS and PIP was 20 m. The Parsivel was located inside of the double fence intercomparison reference, which was located 35 m from VISSS.

### 4.1 Case study comparing VISSS, PIP, and Parsivel

VISSS level2match data are compared with PIP and Parsivel observations for a snowfall case on 26 January 2022. For a fair comparison with PIP and Parsivel that observe particles from a single perspective, only data of a single VISSS camera is used

in this section. Because Parsivel uses something similar to $D_{eq}$ (see discussion in Battaglia et al., 2010, for the predecessor instrument), $D_{eq}$ is also used as a PIP and VISSS size descriptor in the following. Also, $D_{eq}$ is not affected by the problems of the PIP particle sizing algorithm identified by (Helms et al., 2022). The PSD is characterized by the two variables $N_0^*$ and $D_{32}$

used to describe the normalized size distributions $N(D) = N_0^* F(D/D_{32})$ (Testud et al., 2001; Delanoë et al., 2005) where $N_0^*$ is a scaling parameter and $D_{32}$ normalizes the size distribution by size. Assuming a typical value of 2 for the exponent $b$ of the mass-size relation (e.g., Mitchell, 1996), $D_{32}$ is the proxy for the mean mass-weighted diameter defined as the ratio of the third to the second measured PSD moments $M_3/M_2$. Assuming the same value for $b$, $N_0^*$ can be calculated with

$$N_0^* = \frac{M_2^4}{M_3^3} \frac{27}{2} \tag{5}$$

as shown in Maahn et al. (2015). The variability of $N_0^*$ and $D_{32}$ as well as the particle complexity $c$ and the number of particles observed throughout the day are depicted in Fig. 6. The spectral variable $c$ is available for each size bin. Because using a PSD-weighted average over all sizes for $c$ would be heavily weighted to smaller particles which are less complex due to the finite resolution, we use the 95th percentile for $c$ in the following. The main precipitation event lasted from 10:00 to 17:30 UTC and shows an anticorrelation between $N_0^*$ and $D_{32}$: the former increases up to $10^5$ m$^{-3}$ mm$^{-1}$ until 13:00 UTC before decreasing

to $10^3$ m$^{-3}$ mm$^{-1}$ at the end of the event. The particle complexity $c$ divides the core period of the event into two parts with $c \approx$ 2 before 13:00 UTC and $c \approx 2.8$ after 13:00 UTC. This transition can also be seen in the random selection of matched particles observed by the VISSS (Fig. 7) retrieved from the imagesL1detect product. For each particle, a pair of images is available from the two VISSS cameras. Before 13:00 UTC, a wide variety of different particle types has been observed, including plates, small aggregates and small rimed particles. Since particle shape and mean brightness are not used to match particles, the observed

image pairs also confirm the ability of VISSS to correctly match data from the two cameras. After 13:00 UTC, needles and needle aggregates dominate the observations explaining the increase in observed complexity. Towards the end of the event, particles become smaller and more irregularly shaped. Around 18:30 UTC, even some ice lolly shaped particles (Keppas et al., 2017) are observed by the VISSS.

$N_0^*$ and $D_{32}$ are also calculated from the PSDs observed by PIP and Parsivel. For the core event, $N_0^*$ measured by the PIP

is about an order of magnitude smaller than that measured by VISSS and Parsivel. The agreement of VISSS and Parsivel is better, but some peaks in $N_0^*$ are not resolved by the Parsivel when $D_{32}$ is large. This discrepancy may be related to problems of the Parsivel with larger particles reported before (Battaglia et al., 2010). The reason for the observed differences between PIP and VISSS is likely more complex. Overall the measured $D_{32}$ agrees better than $N_0^*$. Because $D_{32}$ is a proxy for the mass-weighted mean diameter, larger more massive snowflakes have a larger impact on $D_{32}$ than more numerous smaller particles.

This implies that PIP is not capturing as many small ice particles as VISSS, while measurements of larger particles seem to be less affected. Tiira et al. (2016) have studied the effect of the left-side PSD truncation on PIP observations (see Fig. 6 in Tiira et al., 2016), but the observed VISSS - PIP difference seems to be somewhat larger than expected, namely the difference extends to larger $D_{32}$ values.

The number of particle observations ranges between 10,000 and 100,000 per minute, showing that estimates of $N_0^*$, $D_{32}$,

and $c$ are based on sufficient number of observations to limit the impact of random errors. This is about 1.5 orders of magnitude

more particles than observed by Parsivel and PIP (Fig. 6.d), but this is not a fair comparison because Parsivel and PIP report the number of unique particles, and the number of particle observations is used here for the VISSS. When applying the tracking algorithm to the VISSS and consider only unique particle observations consistent to the other sensors, the advantage of the VISSS is reduced to 50%. The average track length of the VISSS varies throughout the day between 5 and 20 frames with an overall average of 8.5 frames.

To further investigate the differences between the instruments, we compare VISSS, PIP, and Parsivel PSDs (Fig. 8) for the three discussed times during the snowfall case. While Parsivel and VISSS mostly agree for $D > 1$ mm for all three cases, Parsivel observes more particles for $0.6$ mm $< D < 1$ mm (as previously reported by Battaglia et al., 2010) before dropping for $D < 0.6$ mm, which is likely related to limitations associated with the Parsivel pixel resolution of 125 μm. The comparison of VISSS and PIP shows larger discrepancies as explained above. The PSDs tend to agree for $D_{eq} > 1$ mm for cases where larger ice particles are more spherical (11:24 UTC). For the needle case (13:00 UTC), PIP reports lower number concentrations than VISSS and Parsivel for almost all sizes. At 10:10 UTC, VISSS and PIP approximately agree for sizes between 0.4 and 0.8 mm, but PIP reports lower values for other sizes. Although no needles are observed at 10:10 UTC, Fig. 7 shows that there were also small columns that could be affected by the dilation of structures less than 0.4 mm wide by the PIP software, or some parts of radiating assemblage of plates were removed by the image processing.

All three instruments have different sensitivities to small particles. This can be seen for the drop in $D_{32}$ around 17:45 UTC (Fig. 6) where the Parsivel does not report any values, and the PIP $N_0^*$ estimates differ strongly from the VISSS when $D_{32} < 1$ mm. The VISSS reports $D_{32}$ values as low as 0.16 mm around 19:00 UTC. Although the sample sizes are sufficient ($> 10,000$ particles per minute), the errors are likely large due to the VISSS pixel resolution of ~0.06 mm. In the absence of an instrument designed to observe small particles, it is not possible to determine how reliably VISSS detects and sizes small particles.

Additional insight is provided by comparing the drop size distributions (DSD) observed by the three instruments during a drizzle event on 16 October 2021 (Fig. 8.d). The use of drizzle allows Parsivel to be used as a reference instrument as it has been shown to provide accurate DSDs for sizes between 0.5 and 5 mm (Tokay et al., 2014). In fact, Parsivel and VISSS DSDs differ no more than 10% for $0.55$ mm $> D > 0.9$ mm both showing a dip in the distribution around 0.55 mm. For larger droplets, differences are likely related to their low frequency of occurrence increasing statistical errors. For smaller droplets, VISSS (and PIP) report about an order of magnitude higher concentrations than the Parsivel. Similarly, (Thurai et al., 2019) found that a 50 μm optical array probe observed more small drizzle droplets than a Parsivel. For these small particle sizes close to the VISSS camera pixel resolution, discretization errors likely play a role which we investigate by comparing $D_{max}$ and $D_{eq}$ for the VISSS. As drizzle droplets can be considered sufficiently spherical (i.e. $AR > 0.9$) for $D < 1$ mm (Beard et al., 2010), we can evaluate whether $D_{max} = D_{eq}$ holds as expected (Fig. 8.d). As expected, VISSS $D_{max}$ and $D_{eq}$ are in almost perfect agreement for $D > 0.5$ mm, but larger differences occur for $D < 0.3$ mm indicating that discretization errors can become substantial for $D < 0.3$ mm.

In the absence of a reference instrument for smaller particles in Hyytiälä or reference spheres with diameters smaller than 0.5 mm, the performance of the VISSS for observing small particles with $D < 0.5$ mm is difficult to assess. Particles close to

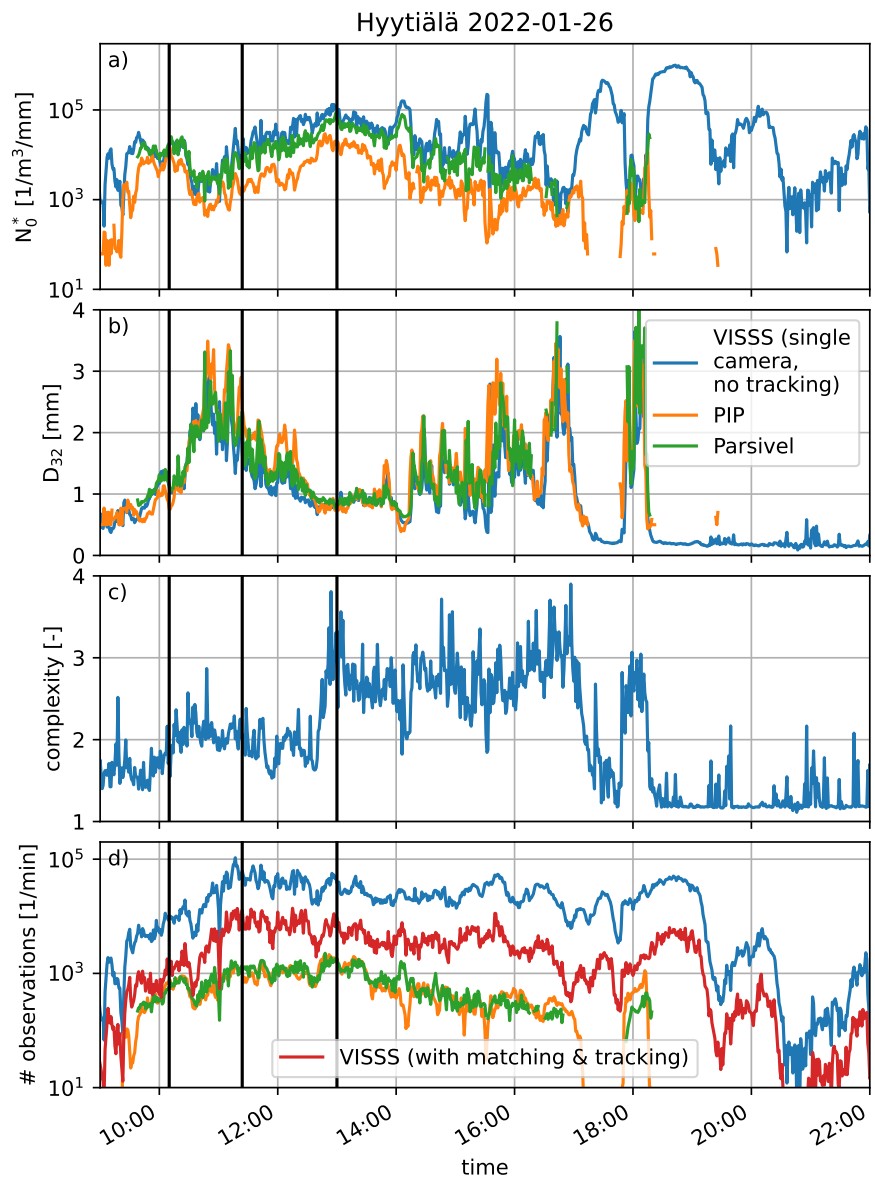

**Figure 6.** Comparison of VISSS (blue), PIP (orange), and Parsivel (green) for a snowfall case on 26 January 2022 at Hyytiälä using $N_0^*$ (a), $D_{23}$ (b), complexity $c$ (c), and the number of observed particles (d). For the VISSS, the latter is shown without (blue) and with (red) particle tracking. The three vertical black lines indicate the sample PSDs shown in Fig. 8.

the thresholds for size, area, and blur might be rejected for parts of the observed trajectory which could explain the decrease in VISSS number concentration for small particle sizes.

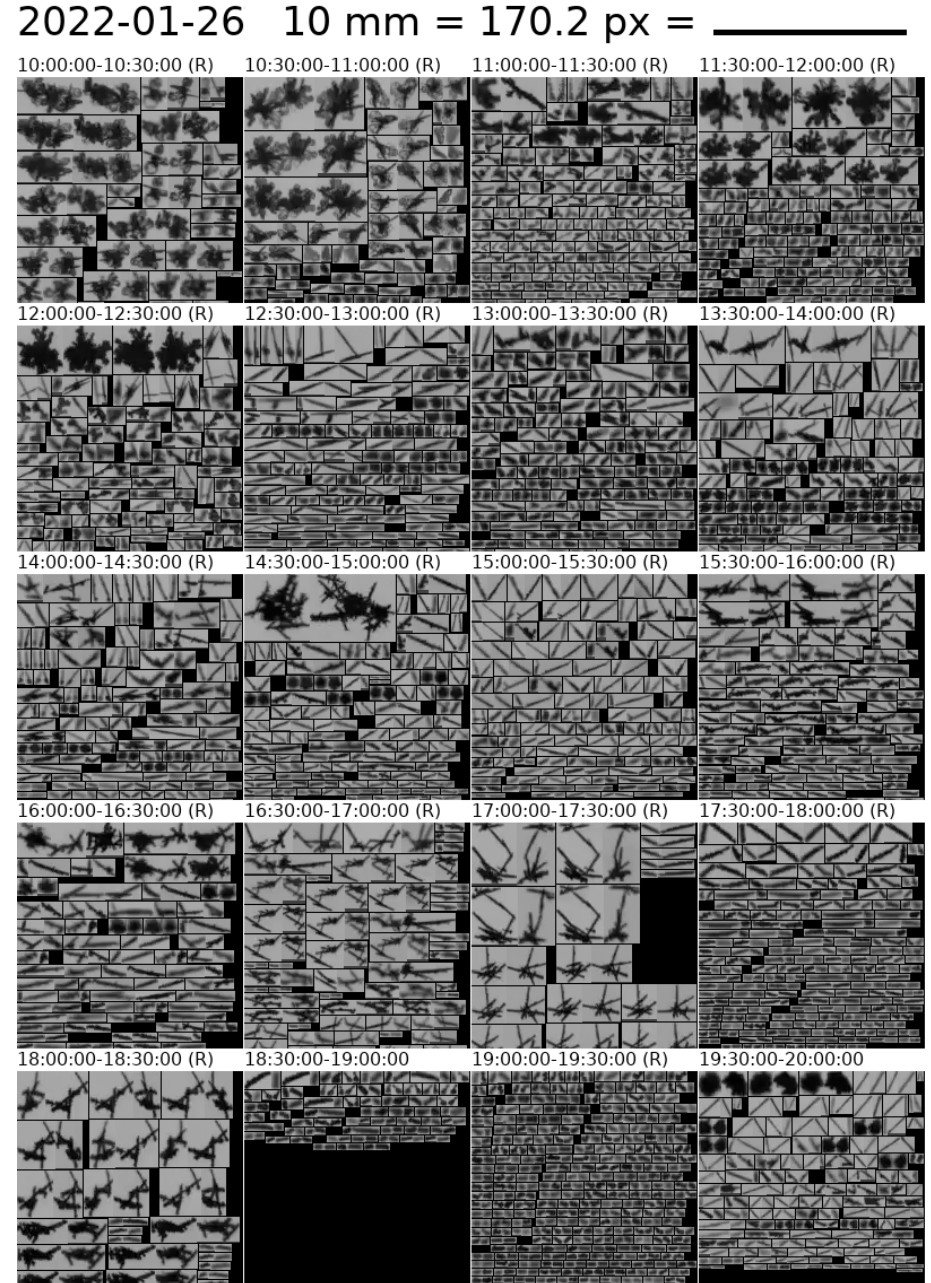

**Figure 7.** Image pairs of particles observed by the two VISSS cameras on 26 January 2022 between 10:00 and 19:00 UTC in original resolution. The (R) indicates that more particles than shown were observed by the VISSS and only a random selection is presented in the panel. Even though particles $\geq 2$ px are processed, only particles with $D_{max} \geq 10$ px (0.59 mm) are shown because the particle shape of smaller particles cannot be identified.

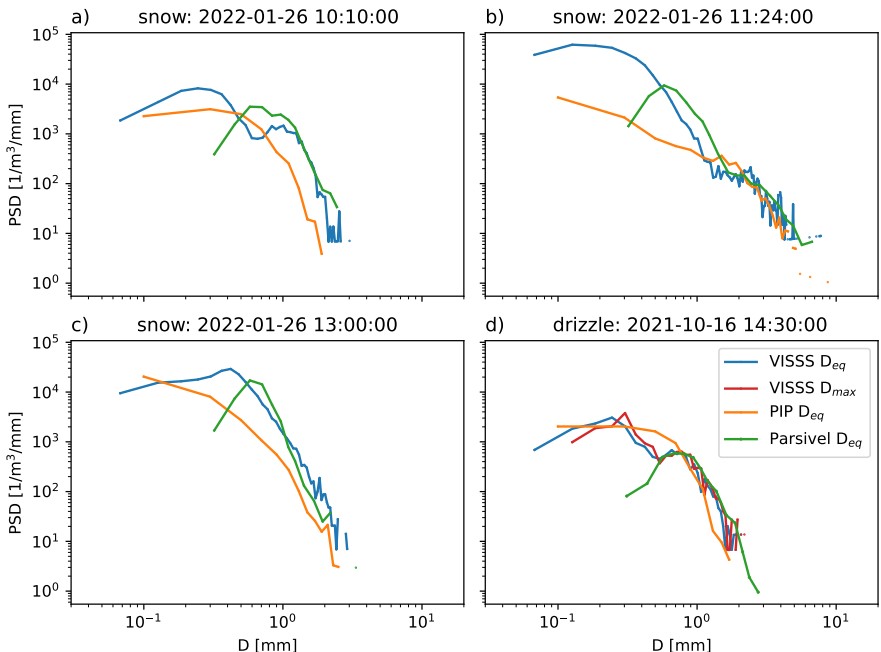

**Figure 8.** (a-c) Particle size distributions of VISSS, PIP, and Parsivel for the three cases indicated in Fig. 6 on 26 January 2022 integrated over 1 minute. $D_{eq}$ is used as a size descriptor. (d) Same as (a-c), but showing the drop size distribution of a drizzle case on 16 October 2021. In addition, the VISSS drop size distribution is also shown with $D_{max}$ as the size descriptor.

### 4.2 Statistical comparison of VISSS, PIP, and Parsivel

The results of the case study comparison of VISSS, PIP, and Parsivel also hold when comparing 6661 minutes of joint snowfall observations during the winter of 2021/22 (Fig. 9). The ratio of $N_0^*$ observed by VISSS and PIP (Parsivel) is compared to $D_{32}$, $N_0^*$, and complexity $c$. For $D_{32} < 1$ mm, the VISSS to PIP (Parsivel) $N_0^*$ ratio increases strongly and can reach a value of 10,000 (10). Therefore, the comparison of the $N_0^*$ ratio with $N_0^*$ itself and $c$ is limited to data with $D_{32} > 1$ mm. For the PIP, the difference in $N_0^*$ does not depend on $N_0^*$ but—as suggested by the needle case above—on complexity $c$, with higher $c$ values indicating larger $N_0^*$ differences, probably related to problems of the PIP image processing. For the VISSS to Parsivel comparison, the $N_0^*$ difference depends rather on $N_0^*$ instead of $c$. Because $D_{32}$ and $N_0^*$ are often anti-correlated, this could be related to size-dependent errors of the Parsivel as identified by Battaglia et al. (2010).

### 4.3 Advantage of the second VISSS camera

Here, we quantify the advantage of observing multiple orientations of a particle with the VISSS. For this, we compare one minute values of mean $D_{max}$, $D_{eq}$, and $p$ obtained from a single camera, using the maximum value obtained from both cameras, and the maximum value obtained during the observed particle track (Fig. 10.a-c). For $AR$, the minimum of the two

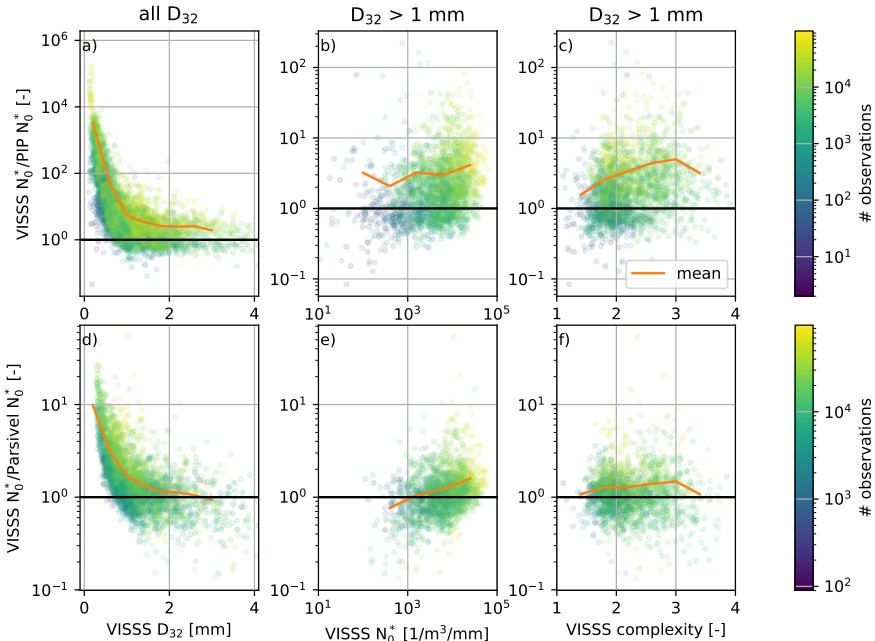

**Figure 9.** Statistical analysis of the ratio of VISSS to PIP $N_0^*$ as a function of (a) VISSS $D_{23}$, (b) VISSS $N_0^*$, and (c) VISSS complexity $c$. (d-f) Same as (a-c), but comparing the VISSS to the Parsivel. The color indicates the number of particles observed by VISSS, the orange line indicates the mean ratio. The analysis for $N_0^*$ (b, e) and $c$ (c, f) is restricted to cases with $D_{23} > 1$ mm.

cameras and along the track is used instead of the maximum (Fig. 10.d). To evaluate the effect of particle type, three cases with mostly dendritic aggregates (6 December 2021, 07:19 - 12:30 UTC), needles (5 January 2022, 00:00 - 14:30 UTC), and graupel (6 December 2021, 00:00 - 04:50; 13:30 - 14:20; 21:15-24:00 and 5 January 2022, 15:00 - 16:40; 19:40 -20:50 UTC) are used. The change in observed values is strongest for needles, which are the most complex particles, where when using two cameras $D_{max}$, $D_{eq}$, $p$, and $AR$ change by 16%, 10%, 14%, and $-12\%$, respectively, and when additionally considering tracking the values change by 24%, 19%, 24%, and $-27\%$, respectively. Changes for dendritic aggregates and graupel are less and surprisingly similar: $D_{max}$ increases by 8% and 7% (13% and 16%), $D_{eq}$ increases by 6% and 6% (14% and 14%), and $p$ increases by 7% and 7% (19% and 16%), respectively, when using two cameras (two cameras with tracking). The dependency of particle properties to orientation can be also seen from the fact that mean $AR$ decreases from 0.62 to 0.54 and 0.42 for aggregates and from 0.73 to 0.67 and 0.54 for graupel highlighting that orientating matters even for graupel.

Underestimating $D_{max}$ can lead to biases when using commonly used $D_{max}$ based power laws for particle mass (Mitchell, 1996) or when using in situ observations to forward model radar observations. This is because scattering properties of non-spherical particles are typically parameterized as a function of $D_{max}$ (Mishchenko et al., 1996; Hogan et al., 2012). Further, particle scattering properties are also impacted by the distribution of particle mass along the path of propagation (Hogan and Westbrook, 2014) which is impacted by $AR$. To analyze how the different $D_{max}$ and $AR$ estimates affects the simulated

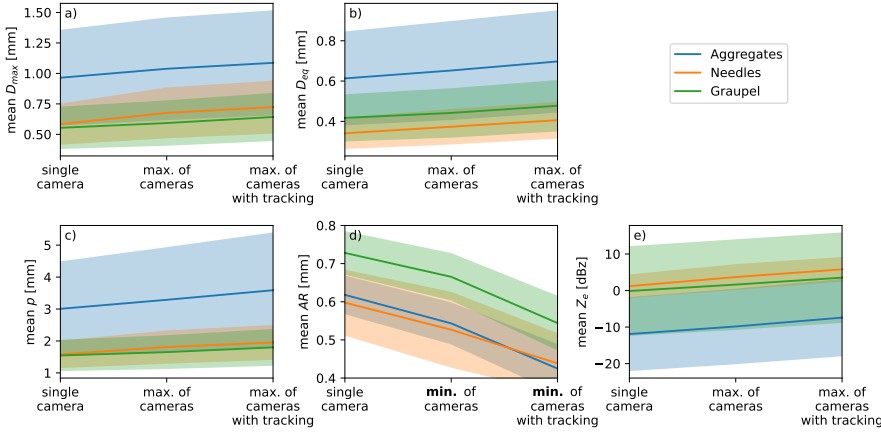

**Figure 10.** Mean (a) $D_{max}$, (b) $D_{eq}$, (c) perimeter $p$, (d) aspect ratio $AR$, and (e) radar reflectivity factor $Z_e$ for cases with aggregates (blue), needles (orange), and graupel (green) when using only a single VISSS camera, both cameras, and both cameras considering particle tracking. The shaded areas denote one standard deviation.

radar reflectivity for vertically pointing cloud radar observations at 94 GHz, we use the PAMTRA radar simulator (Passive and Active Microwave radiative TRAnsfer tool, Mech et al., 2020) with the riming-dependent parameterization of the particle scattering properties (Maherndl et al., 2023a) assuming horizontal particle orientation (Sassen, 1977; Hogan et al., 2002). Using two cameras (i.e., max($D_{max}$, min($AR$)) increases mean $Z_e$ values by 2.1, 2.5 and 1.8 dB for aggregates, needles, and graupel, respectively. When exploiting also the varying orientations during tracking, the offsets increase to 4.5, 4.6, and 3.7 dB, respectively, which is considerably larger than the commonly used measurement uncertainty of 1 dB for cloud radars. The change in $Z_e$ is similar to the 3.2 dB found by Wood et al. (2013) using idealized particles.

## 5  Conclusions

The hardware and data processing of the open source Video In Situ Snowfall Sensor (VISSS) has been introduced. The VISSS consists of two cameras with telecentric lenses oriented at a 90° angle to each other and observe a common observation volume. Both cameras are illuminated by LED backlights (see Table 1 for specifications). The goal of the VISSS design was to combine a large, well defined observation volume and relatively high pixel resolution with a design that limits wind disturbance and allows accurate sizing. The VISSS was initially developed for MOSAiC, but additional deployments at Hyytiälä, Finland and Gothic, Colorado USA followed. An advanced version of the instrument has been installed at Ny-Ålesund, Svalbard. The VISSS level 1 processing steps for obtaining per-particle properties include particle detection and sizing, particle matching between the two cameras considering the exact alignment of the cameras to each other, and tracking of individual particles to estimate sedimentation velocity and improve particle property estimates. For level 2 products, the temporally averaged particle properties and size distributions are available in calibrated metric units.

The initial analysis shows the potential of the instrument. The relatively large observation volume of the VISSS leads to robust statistics based on up to 10,000 individual particle observations per minute. The data set from Hyytiälä obtained in the winter of 2021/22 is used to compare the VISSS with collocated PIP and Parsivel instruments. While the comparison with the Parsivel shows—given the known limitations of the instrument for snowfall (Battaglia et al., 2010)—excellent agreement, the comparison with the PIP is more complicated. The differences in the observed PSDs increase with increasing particle complexity $c$ (e.g., needles), but differences remain even for non-needle cases and for a case with a relatively high concentration of large, relatively spherical particles, agreement was only found for sizes larger than 1 mm. Because the Parsivel is well characterized for liquid precipitation (Tokay et al., 2014), a drizzle case is also used for comparison. The case shows an excellent agreement between Parsivel and VISSS for droplets larger than 0.5 mm, confirming the general accuracy of VISSS. Compared to both PIP and Parsivel, VISSS observes a larger number of small particles that can drastically change the retrieved PSD coefficients in some cases. However, the first generation VISSS pixel resolution of 0.06 mm is likely to introduce discretization errors for particles smaller than 0.3 mm (i.e. 5 px), potentially leading to errors in the sizing of very small particles. Furthermore, we analyzed the advantage of the VISSS due to the availability of a second camera. Depending on the particle type, mean $D_{max}$ increases up to 16% and mean aspect ratio $AR$ decreases by 12%. For the analyzed case, the VISSS observes each particle on average 8.5 times which can further improve estimates of particle properties due to the natural rotation of the particle during sedimentation. In comparison to using only a single camera, this can increase mean $D_{max}$ by up to 24% and reduce $AR$ by up to 31%.

VISSS product development will continue, e.g., by implementing machine learning based particle classifications (Praz et al., 2017; Leinonen and Berne, 2020; Leinonen et al., 2021). Also, we will work on making VISSS data acquisition and processing more efficient by handling some processing steps on the data acquisition system in real-time. We invite also the community to contribute to the development of the open source instrument. This applies not only to the software products, but allows also for other groups to build and improve the instrument. It could even mean to advance the VISSS hardware concept further, by e.g. adding a third camera to observe snow particles from below or—given the extended 1300 mm working distance of VISSS3—from above. The VISSS hardware plans (2nd generation VISSS, Maahn et al., 2023), data acquisition software (Maahn, 2023a), and data processing libraries (Maahn, 2023b) have been released under an open source license so that reverse engineering as done by Helms et al. (2022) is not required to analyze the VISSS data processing. The only limitation of the used licenses is that modification of the VISSS need to be made publicly available under the same license. Hardware plans for the third VISSS generation will be published on completion of the instrument end of the year.

There are many potential applications for VISSS observations. It can be used for model evaluation with advanced microphysics (e.g., Hashino and Tripoli, 2011; Milbrandt and Morrison, 2015), characterization of PSDs as a function of snowfall formation processes, or retrievals combining in situ and remote sensing observations. Tracking of a particle in three dimensions can be used to understand the impact of turbulence on particle trajectories. Beyond atmospheric science, the VISSS shows potential for quantifying the occurrence of flying insects, as standard insect counting techniques such as suction traps are typically destructive and labor-intensive.

*Code and data availability.* VISSS hardware plans (Maahn et al., 2023), data acquisition software (Maahn, 2023a), and data processing libraries (Maahn, 2023b) have been released under an open source license. VISSS, PIP, and Parsivel observations used for the pilot study are available at https://zenodo.org/record/7797286 (Maahn and Moisseev, 2023).

## Appendix A:  Coordinate system transformation

We use a right handed coordinate system $(x,y,z)$ to define the position of particles in the observation volume, where $z$ points to the ground (see Fig. 1). The follower coordinate system $(x_F,y_F,z_F)$ can be transformed into the leader coordinate system $(x_L,y_L,z_L)$ by the standard transformation matrix

$$\begin{pmatrix} x_L \\ y_L \\ z_L \end{pmatrix} = \begin{pmatrix} \cos\theta\cos\psi & \sin\varphi\sin\theta\cos\psi - \cos\varphi\sin\psi & \cos\varphi\sin\theta\cos\psi + \sin\varphi\sin\psi \\ \cos\theta\sin\psi & \sin\varphi\sin\theta\sin\psi + \cos\varphi\cos\psi & \cos\varphi\sin\theta\sin\psi - \sin\varphi\cos\psi \\ -\sin\theta & \sin\varphi\cos\theta & \cos\varphi\cos\theta \end{pmatrix} \begin{pmatrix} x'_F \\ y'_F \\ z'_F \end{pmatrix} \tag{A1}$$

using the follower's roll $\varphi$, yaw $\psi$, and pitch $\theta$, analogous to airborne measurements, and with $x'_F = x_F + O_{fx}$, $y'_F = y_F + O_{fy}$, and $z'_F = z_F + O_{fz}$, where $O_{fx}$, $O_{fy}$, and $O_{fz}$ are the offsets of the follower coordinate system in the $x$, $y$, and $z$ directions, respectively (see Fig. 1) Offsets in $O_{fx}$ and $O_{fy}$ are neglected, because they would only materialize in reduced particle sharpness, but not in the retrieved three-dimensional position. The opposite transformation can be described by:

$$\begin{pmatrix} x'_F \\ y'_F \\ z'_F \end{pmatrix} = \begin{pmatrix} \cos\theta\cos\psi & \cos\theta\sin\psi & -\sin\theta \\ \sin\varphi\sin\theta\cos\psi - \cos\varphi\sin\psi & \sin\varphi\sin\theta\sin\psi + \cos\varphi\cos\psi & \sin\varphi\cos\theta \\ \cos\varphi\sin\theta\cos\psi + \sin\varphi\sin\psi & \cos\varphi\sin\theta\sin\psi - \sin\varphi\cos\psi & \cos\varphi\cos\theta \end{pmatrix} \begin{pmatrix} x_L \\ y_L \\ z_L \end{pmatrix} \tag{A2}$$

Since we have only one measurement in the $x$ and $y$ dimensions, but two in $z$, we use the difference between the measured $z_L$ and the estimated $z_L$ from matched particles to retrieve the misalignment angles and offsets

$$z_L = -\sin\theta x'_F + \sin\varphi\cos\theta y'_F + \cos\varphi\cos\theta z'_F. \tag{A3}$$

In this equation, $x'_F$ is unknown so it is derived from

$$x'_F = \cos\theta\cos\psi x_L + \cos\theta\sin\psi y_L - \sin\theta z_L \tag{A4}$$

where, in turn $y_L$ is not observed. Therefore, $y_L$ is obtained from

$$y_L = \cos\theta\sin\psi x'_F + (\sin\varphi\sin\theta\sin\psi + \cos\varphi\cos\psi)y'_F + (\cos\varphi\sin\theta\sin\psi - \sin\varphi\cos\psi)z'_F. \tag{A5}$$

Inserting equations A5 into A4 yields after a couple of simplifications

$$x'_F = \frac{\cos\theta\cos\psi}{1-\cos^2\theta\sin^2\psi}x_L$$
$$+ \frac{(\cos\theta\sin\varphi\sin\theta\sin^2\psi + \cos\varphi\cos\psi\cos\theta\sin\psi)}{1-\cos^2\theta\sin^2\psi}y'_F$$
$$+ \frac{(\cos\theta\cos\varphi\sin\theta\sin^2\psi - \sin\varphi\cos\psi\cos\theta\sin\psi)}{1-\cos^2\theta\sin^2\psi}z'_F$$
$$- \frac{\sin\theta}{1-\cos^2\theta\sin^2\psi}z_L. \tag{A6}$$

Inserting equations A6 into A3 yields:

$$z_L = -\frac{\sin\theta}{\cos\theta\cos\psi}x_L$$
$$- \frac{\sin\theta\sin\psi\cos\varphi - \cos\psi\sin\varphi}{\cos\theta\cos\psi}y'_F$$
$$+ \frac{\sin\theta\sin\psi\sin\varphi + \cos\psi\cos\varphi}{\cos\theta\cos\psi}z'_F. \tag{A7}$$

We have no information about $\psi$, therefore we have no choice but assuming $\psi = 0$ leading to

$$z_L = -\frac{\sin\theta}{\cos\theta}x_L + \frac{\sin\varphi}{\cos\theta}y'_F + \frac{\cos\varphi}{\cos\theta}z'_F. \tag{A8}$$

*Author contributions.* MM acquired funding, developed the instrument, processed the VISSS data, analyzed the data of the case study, and wrote the manuscript. DM processed PIP and Parsivel data and contributed to data analysis. NM and IS contributed to instrument calibration and particle tracking development, respectively. MS supported funding acquisition and was responsible for the VISSS deployment at MOSAiC. All authors reviewed and edited the draft.

*Competing interests.* MM is a member of the editorial board of Atmospheric Measurement Techniques but was not involved in the peer-review process of this paper.

*Acknowledgements.* Funded by the German Research Foundation (DFG, Deutsche Forschungsgemeinschaft) Transregional Collaborative Research Center SFB/TRR 172 (Project-ID 268020496), DFG Priority Program SPP2115 "Fusion of Radar Polarimetry and Numerical Atmospheric Modelling Towards an Improved Understanding of Cloud and Precipitation Processes" (PROM) under grant PROM-CORSIPP
(Project-ID 408008112), and the University of Colorado Boulder CIRES (Cooperative Institute for Research in Environmental Sciences) Innovative Research Program. MDS was supported by the National Science Foundation (OPP-1724551) and National Oceanic and Atmospheric Administration (NA22OAR4320151). The deployment at Hyytiälä was supported by an ACTRIS-2 TNA funded by the European

Commission under the Horizon 2020 – Research and Innovation Framework Programme, H2020-INFRADEV-2019-2, Grant Agreement number: 871115. The PIP deployment at the University of Helsinki station is supported by the NASA Global Precipitation Measurement Mission ground validation program. We thank all persons involved in the MOSAiC expedition (MOSAiC20192020) of the Research Vessel Polarstern during MOSAiC in 2019–2020 (Project ID: AWI_PS122_00) as listed in Nixdorf et al. (2021), in particular Christopher Cox, Michael Gallagher, Jenny Hutchings, and Taneil Uttal. In Hyytiälä, the VISSS was taken care of by Lauri Ahonen, Matti Leskinen, and Anna Trosits. In Ny-Ålesund, the VISSS installation was made possible by the AWIPEV team including Guillaume Hérment, Fieke Rader, and Wilfried Ruhe. During SAIL, we were supported by the Operations team from Rocky Mountain Biological Laboratory team and the DOE Atmospheric Radiation Measurement technicians who took great care of the VISSS. Thanks to Donald David, Rainer Haseneder-Lind, Jim Kastengren, Pavel Krobot, and Steffen Wolters for assembling VISSS instruments. We thank Thomas Kuhn and Charles Helms for their extensive and constructive reviews.

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
