# Peer review of "Introducing the Video In Situ Snowfall Sensor (VISSS)"

_EGUsphere, 2023_

## Referee Comment (RC2)

Review of Introducing the Video In Situ Snowfall Sensor (VISSS)
Reviewed by Charles Helms

This manuscript describes the VISSS instrument. VISSS is a new video-based precipitation microphysics probe designed to capture high-resolution images of snowflakes using a pair of orthogonally-pointing high-speed cameras. The manuscript also compares the VISSS measurements to those of two other precipitation microphysics probes: PIP and Parsivel. Overall, I found the manuscript to be of high quality, although there are some minor improvements that would further improve upon this quality.

As I only feel minor revisions are necessary, I've opted to put all my comments (some of which are simply small typo corrections) in the order they appear in the text.

Line 6 (also Lines 317 and 409): Is VISSS observing up to 100,000 unique particles per minute (i.e., a falling particle is only counted once during its transit across the domain) or is it making up to 100,000 particle observations per minute (i.e., each measurement of a particle is counted, even if that particle has been measured in a previous frame)? If it's the latter, I suggest changing the wording to "100,000 particle observations per minute"

Line 6: This is the first time PIP is mentioned; suggest moving the definition of the PIP acronym from Line 8 to here

Line 52 (also Line 378): The Del Guasta (2022) reference is inserted parenthetically here instead of being in-line (similar for the Battaglia et al. reference on line 378).

Line 57: The PIP acronym was already defined above, although, personally, I don't see any problem with it being defined a second time so I leave changing this up to author discretion.

Line 60 (and elsewhere): Maybe a bit pedantic, but 100 microns per pixel is the pixel size rather than the resolution (i.e., the minimum resolvable particle size).

Paragraph starting on Line 89: I really appreciate that the authors include the details about the camera in the text (and in table 1). The only additional piece of information the authors might consider adding is the type of camera (i.e., CCD, CMOS Global Shutter, CMOS Rolling Shutter, etc).

Line 113: suggest removing the word "also" to improve readability (authors' discretion)

Table 1: The pixel sizes are inconsistent in their use of "." or "," as a decimal point.

Table 1: If there is room for it in the table, I suggest changing "Exposure time" to "Effective exposure time" to make it clear that this is the duration of the LED being on rather than the actual exposure time of the camera itself

Figure 3: It looks like the blue ellipse only appears in the final annotated image. If this is intentional, it might be worth adding a note that the fitEllipse shape is only annotated on that image otherwise it would be helpful to either note that the blue ellipse is obscured by another ellipse (and indicate which ellipse this is) or use a dashed line for the blue ellipse and put it on top of the other ellipses.

Line 159: "sphere" should be changed to "circle"

Line 175: Should "vertical position" be "horizontal position"?  I would think the vertical position information would be known from the leader camera.  If not, this discrepancy might need a sentence or two of brief explanation.

Paragraph starting on Line 183:  It's still not clear to me how the frame matching works.  Is this a case of matching up the first frame that the particle appears in?  If so, wouldn't this require the alignment of the two cameras to be extremely good (on the order of half a pixel or less, presumably)?  Or is this more manually intensive than I'm thinking it is and the matching is based on matching up how the particle tumbles as it passes through the domain?  Regardless, more details would be helpful.  (A side thought that occurred to me while reading this: have you considered matching the frames up by using a camera flash while the LEDs are obscured?  I'm not sure if that'll work or not, but figured I'd mention it anyway)

Line 213: "reader" should be "leader"

Section 3.4: I suggest removing the proof of concept tracking (and the related figure 4).  It doesn't really add much to the manuscript and nearest-neighbor-based particle tracking is really bad outside of very low winds and/or very light snowfall.  Depending on how much progress has been made, it might be worth adding some more information on the actual particle tracking algorithm the authors are developing.

Line 269 – 271: If multiple observations of a single particle are included in the PSD, wouldn't this bias the PSD towards slower falling particles?

Line 313: It took me several times reading these sentences to realize the "Fig 6. *c*" did not refer to panel c of figure 6, but that these are two completely separate thoughts; suggest replacing "c" with "Particle complexity" or "Particle complexity *c*" to avoid this issue.

Line 314: insert "particles" after "weighted to smaller"

Lines 336 – 354 (regarding the PIP underestimation of N0*):  I don't think the dilation is the issue here.  PIP's processing applies an edge detection filter, dilates the resulting image twice (using the 3x3 kernel), fills in any holes, and then erodes the hole-filled image twice to (theoretically) undo the dilation step (using the same 3x3 kernel).  A few possibilities that come to my mind to explain the discrepancies between VISSS and PIP are: 1) the dilation is merging nearby needles into a single particle, thus decreasing the number of small particles (presumably this would be paired with an increase in larger particles); 2) the image compression (which averages vertically-adjacent pixels to reduce the data rates) is essentially destroying the smaller needles; and 3) that the dilation and hole-filling of higher complexity particles is artificially inflating the equivalent diameter and this is introducing a bias into the PSD moments used to compute N0* (presumably this one is less relevant for the needles, but later you mention PIP having issues with high complexity particles, so I included this as well).  Unfortunately, I don't have any deeper insight as to which one of these might be the culprit (if it even is one of these), but I'll add this to the top of my list of PIP behaviors to look into.  Either way, I don't think it would be the dilation itself as that adds to the particles.

From the VISSS side of things, if the PSD is being biased towards slower falling particles by including all particle observations, this might produce a bias at small particles in the VISSS PSD.  It should be relatively easy to test this by comparing the VISSS PSD as it appears here to the PSD that would be produced if only particles appearing in every 25[th] frame are included in the PSD (this is how PIP

computes its PSD, for reference).  Under normal conditions, 25 frames should be more than enough time for any particles observed in a frame to exit the measurement volume.

Line 337: should "width" be "length"?  In my mind the width of a particle is more closely aligned with (one of) the shorter axis of a particle, but when reading this it feels like the authors are referring to the longer axis of the needles.

Lines 342 – 343 and 352 – 353: As I mentioned above, dilation shouldn't result in the removal of any parts of a snowflake as dilation expands the particles.  Erosion could, in theory, remove parts, but I doubt it since the erosion step occurs after the particles have already been dilated twice.

Fig. 6: It might be helpful to also include the observed particles per minute for the PIP and Parsivel just to give a point of comparison.  For PIP at least, the PSD is computed using the *_a_p_60.dat files (which only includes particles observed during every 25$^{th}$ frame to avoid double counting).

Additionally, if the authors' wish to, the particle complexity can be computed from the PIP files by dividing the particle area by the hydraulic radius (Hy_Rad), which is the ratio between area and perimeter to get the particle perimeter and then plugging the relevant values into Eq 1.  That said, as discussed in Helms et al. (2022) [section 3.1], the PIP software takes some potentially questionable (when applied to a snowflake) shortcuts when computing the perimeter, so comparing particle complexity between VISSS and PIP may not be particularly informative as to the accuracy of VISSS.  I am less familiar with the specifics of the Parsivel output files, but similar methods may be possible there as well.   Either way, I leave choice of adding this up to the authors' discretion.

Fig. 7: It's hard to make anything out on the image due to the small size of each frame.  If might be beneficial to include a subset of these to make each frame larger so readers can better appreciate the resolution of the cameras.

Fig. 8: It would be helpful to have the time period over which the PSDs are computed for each of the instruments either in the caption and/or the text (apologies if this is in the text and I missed it, I added this comment after having read through the paper and didn't see it mentioned when looking back through again).

Line 432: I certainly appreciate VISSS being open source!  We (collectively) shouldn't have to pull teeth to understand how instruments produce their measurements.

Appendix A: It took a couple attempts, but I was able to replicate the derivation in Appendix A.

---

## Author Comment (AC1)

**Introducing the Video In Situ Snowfall Sensor (VISSS)**
**Response to the reviewers**

Maximilian Maahn, Dmitri Moisseev, Isabelle Steinke,
Nina Maherndl, and Matthew D. Shupe

September 27, 2023

*Original Referee comments are in italic*

> manuscript text is indented, with added text underlined and

We would like to thank the reviewers for their very helpful comments. We revised the manuscript and responded to all of the reviewers' comments.

Besides addressing the reviewers' comments, we also included a description of the new tracking algorithm to the manuscript.

**1 Review by Thomas Kuhn**

*The manuscript describes a new instrument to image individual snowflakes. It represents a relevant and useful contribution to the relatively few instruments that image snowflakes and collect detailed information on snowfall in this way. VISSS, the new instrument, is in its working principle similar to the SVI/PIP as they both use video imaging of a relatively large sampling volume ( 5cm x 5cm x 5cm) with illumination from the back. The VISSS has an improved resolution as well as better optics to minimizing sizing errors. The VISSS is different from SVI/PIP as it uses two video cameras with orthogonal viewing directions. The 2-DVD uses already a similar approach, however, with lower-resolution line cameras and issues when reconstructing images from the recorded lines. Thus, the VISSS provides more reliable data. The two viewing directions of the VISSS allow to properly define the sampling volume independent of the imaged particle's size.*

*This is an important advantage of the VISSS. In addition, the two views provide of course more information on each single particle. Even more information can be derived from the multiple exposures of the same snow particle as it is falling through the sampling volume, for example the fall speed.*

*I am complementing the authors to their open approach publishing all design and software.*

*I recommend publication of this manuscript after a minor revision that should address a few questions and issues that I am describing below. I am first raising a few important points and then give feedback on other minor things or suggest corrections.*

We thank Thomas Kuhn for the extensive review and very constructive comments.

**Important points – specific comments**

**1) Resolution**

*When talking about "resolution" (e.g. L 74 "resolution of 43 to 59 µm/px") you almost exclusively refer to what I would call "pixel resolution", i.e. what size on the object does one pixel on the image correspond to. To properly characterize an instruments capability to resolve fine details one should give both the pixel resolution as well as the actual optical resolution that is realized with the imaging system. Optical resolution may be defined and measured in several ways, but I would propose to simply state (and show with examples) what the finest detail is that can be resolved. Even if the optical resolution of the optics may be better, I doubt that the finest detail that can be resolved is on the order of one pixel. Looking at example images in Fig.3, I would estimate the finest detail that can be resolved to be on the order of 100 µm.*

see below.

*In L 112 "quality of the lens proved to be borderline for the applications, resulting in slightly blurred particle images" you touch on optical resolution.*

Thanks for pointing this out, we use the term pixel resolution consistently now. We tested the actual resolution with a microscope lens with 100 µm ticks resolution and we would say the results are generally consistent with the resolution of 58.75 µm and 43.125 µm for the VISSS1 and VISSS2, respectively (see Fig. R.1). However, we would have expected slightly clearer tick marks for the VISSS2 so the optical resolution is likely in the order of 50 µm. We modified:

> However, the optical quality of the lens proved to be borderline for the applications, resulting in an estimated optical resolution of approximately 50 µm

[Figure]

Figure R.1: Microscope scale observation with VISSS1 (left, on 21-09-07) and VISSS2 (right, on 21-10-09).

and slightly blurred particle images, . Consequently, the lens was changed again for the third generation VISSS3 (currently under construction), which  has a working distance of 1300 mm.

**2) Calibration**

*The calibration you present in Sect 3.6 compares the diameter in pixels determined using the image processing (Sect.3.1) to the actual diameter of reference spheres. The slope of the fitted relationship shown in Eq. 5 (or 6) corresponds to the pixel resolution (which you confirmed with an imaged millimeter scale). The interesting result of the calibration (that you cannot get from the millimeter scale) is the offset in Eq. 5. Of course, you should then use the inverse of Eq.5 to convert determined size in pixels to µm. Then, whatever caused the offset will be taken care of. I am wondering why a similar calibration is not done for area and perimeter. You could determine, similar to Eq.5, relationships between determined properties in pixels and actual properties of the reference sphere. This would account for certain effects of the image processing (at least for spheres). I think this would be more accurate than simply using the slope only for converting areas and perimeters.*

see below.

*I don't agree with your explanation of the offset. L 283 "the $D_{max}$ estimator used to process the images often rounds up to the next full pixel" sounds difficult to believe. If*

*this is true, then I highly recommend that you change the $D_{max}$ estimator function. I expect that the offset, at least in part, is due to image processing. After all processing steps (Gaussian blur, Canny filter, dilation, finding the contour, filling and eroding the contour) the resulting size may be offset with a certain bias. I would be curious what would happen to arteficial particle images during processing. Take for example a 2px by 2px square (which should have a $D_{max}$ of about 2.8px, cross-sectional area of 4px, and perimeter of 8px) and see what properties will be determined. If you would do this for a few sizes, you could find a relationship as in Eq.5.*

Thanks for pointing this out, the explanation of the offset was indeed not correct. We followed the suggestion of using synthetic observations to get further insights into the calibration and rewrote the whole calibration section:

> Calibration is required to convert $D_{max}$, $D_{eq}$, and $p$ from pixels to µm. It depends not only on the optical properties of the lens but also on the used computer vision routines. Calibration is obtained using reference steel or ceramic spheres with 1 to 3 mm diameter that are dropped into the VISSS observation volume. After processing using the standard VISSS routines, the estimated sizes are compared to the expected ones. A linear least square fit is applied to the  604 reference sphere observations obtained at Hyytiälä and SAIL resulting in

$$D\underline{px}_{max}[px] = (\underline{0.016971}0.01700 \pm \underline{0.000015}.00001) \cdot D\underline{um}_{max}[um] + (\underline{0.349303}0.49301 \pm \underline{0.02717}0.0210 \tag{1}$$

> for the VISSS1 (Fig. 5.a) and

$$D\underline{px}_{max}[px] = (\underline{0.023047}0.02311 \pm \underline{0.000050}.00003) \cdot D\underline{um}_{max}[um] + (\underline{0.900593}0.81569 \pm \underline{0.078123}.0699 \tag{2}$$

> for the VISSS2 based on  372 samples from Ny-Ålesund (Fig. 5.b). The inverse of the slope is  58.832 µm px$^{-1}$ ( 43.266 µm px$^{-1}$) and is close to the manufacturer's specification of 58.75 µm px$^{-1}$ (43.125 µm px$^{-1}$) for the VISSS1 (VISSS2). The random error estimated from the normalized root mean square error obtained from the difference between observed and expected size is less than 0.8% indicating that random errors are negligible. To investigate the source of the non-zero intercept , we also tested the VISSS computer vision routines with artificially created VISSS images with drawn spheres and compared the expected to measured $D_{max}$

 by a least squares fit (Fig. 5.c). Gaussian blur with a standard deviation between 0 and 3 px was applied to account for a realistic range of blurring due to e.g., motion blur or particles that are slightly out of focus. Note that in addition to that a Gaussian blur filter with a standard deviation of 1.5 px needs to be applied during image processing for the Canny edge detection as discussed above. For the artificial spheres, the obtained slope deviates less than 2% from the expected slope of 1.0, but the offset ranges from 0.6 to 1.5 px caused by the seeming enlargement of the particle due to the applied blur. To investigate the shape dependency of the results, we repeated the experiment with squares (Fig. 5.d). Again, the slope deviates less than 2% from 1.0, but the offset is this time negative with values ranging between $-1.4$ px and $-2.9$ px depending on blur. This is because the corners of the square are rounded when applying Gaussian blur so that the true $D_{max}$  can no longer be obtained. In summary, the VISSS routines overestimate $D_{max}$ of spheres, but underestimate $D_{max}$ of squares. In reality, the VISSS observes a wide range of different shapes that can be both rather spherical or rather complex with "pointy" corners. Therefore, we decided to set the intercept to 0 when calibrating $D_{max}$ which can cause a particle shape dependent bias of $\pm 4$ to  $\pm 6\%$. For particles smaller than 10 px, this bias can be slightly larger due to discretization errors as can be seen from the larger impact of blur for small squares (Fig. 5.d).

For better comparison with $D_{max}$,  $D_{eq}$ is used instead of $A$ for testing the computer vision method for estimating $A$ (Fig 5.e-h). The results are almost identical to $D_{max}$ so that the slopes derived from $D_{max}$ are applied to $D_{eq}$ (and consequently $A$) as well.

For the perimeter $p$ (Fig. 5.i-l), the slopes derived from the reference spheres are about 5% steeper than for $D_{max}$ indicating that VISSS $p$ are biased high. This bias is also found for artificial spheres independent of the applied additional blur. Therefore, this bias is related to the image processing and most likely caused by the Gaussian blur required for the Canny edge detection. For squares, however, the slope is close to 1 likely due to compensating effects caused by "cutting corners" of the algorithm. In reality, the VISSS observes more complex particles for which the perimeter increases with decreasing scale. (compare to coast line paradox, Mandelbrot, 1967). Therefore, we conclude that it is extremely unlikely that the perimeter of real particles is biased high like for artificial spheres but rather biased low depending on complexity. As a pragmatic approach, we also apply the $D_{max}$ slope to $p$ but stress that $p$ has a considerably higher uncertainty than $D_{max}$ or $D_{eq}$.

**3) Smallest particle**

*It would be interesting to know what the smallest particles are that can be measured (or are considered). I have read somewhere a condition of >= 2px for size and >=2px for area. I am not sure that 2px is really a meaningful limit. This is related to my comments on calibration and resolution above. After imposing your 2px-conditions, do you actually observe 2px-particles? If yes, did you examine them by looking at the actual images compared to the contour? If you took a 2px artificial particle, what size and area would be determined by image processing? After such testing, could you state what the smallest particles are that can be measured?*

The 2 px for $D_{max}$ and area limit was motivated by avoiding that the particle detection picks up noise as particles and is actually quite conservative. We looked extensively at individual particles and the results of the particle detection and also small particles are correctly sized. This can be also seen from the size distributions in Fig. 8. The left most data point for $D_{max}$ in Fig. 8.d corresponds to the size class 2 to 3 pixels. When using $D_{eq}$, there is even data in the size class 1 to 2 pixels because an area of 2 px corresponds to an $D_{eq}$ of 1.6 px. We are actually more concerned whether all these small particles are actually all detected or whether some of them are skipped because the thresholds of $D_{max}$, area, and blur are not reached for every frame. We added:

> In the absence of a reference instrument for smaller particles in Hyytiälä or reference spheres with diameters smaller than 0.5 mm, the performance of the VISSS for observing small particles with $D <$0.5 mm is difficult to assess. Particles close to the thresholds for size, area, and blur might be rejected for parts of the observed trajectory which could explain the decrease in VISSS number concentration for small particle sizes.

See above for discussion of artificial particles.

*In Fig 7 you show only particles larger than about 10px. Is this the smallest particle?*

No, the 10 px threshold is only for plotting because the shape of smaller particles cannot be recognized anyhow. We moved this information to the caption to give context.

> Even though particles $\geq$ 2 px are processed, only particles with $D_{max} \geq$ 10 px (0.59 mm) are shown because the particle shape of smaller particles cannot be identified.

**4) Sizing errors**

*Apart from Eq. 5, you don't seem to estimate an error in sizing. In addition to the uncertainties captured by the calibration (Eq. 5), I would expect image blur to cause an*

*error. Particles moving at typical fall speeds, may blur during the 60 μs exposure time by about 1px. This may introduce an additional error. Was calibration done with moving or stationary reference spheres? As a result, calibration may or may not account for motion blur. All image processing related effects should be accounted for by calibration, then the error related to these effects could be less than 0.1px given the uncertainties in offset in Eq.s 5 and 6. It would be could to briefly discuss sizing errors, speculate about motion blur or potentially other error sources (e.g. can we assume that the telecentric lenses completely eliminate sizing errors), and give an estimated error (perhaps depending on size). Can this error be smaller than the finest detail that can be resolved (optical resolution of your imaging system, see my comments on Resolution above)?*

Yes, calibration was done with a moving sphere so all image processing related effects should be accounted for to the extent this is possible when using spheres. We extended:

Calibration is obtained using reference steel or ceramic spheres with 1 to 3 mm diameter that are dropped into the VISSS observation volume.

Regarding blur we added

Image quality is potentially also impacted by motion blur and the exposure time of 60 μs was selected to limit motion blur of particles falling at 1 m/s to 1.02 and 1.44 pixels for VISSS1 and VISSS2, respectively. Particle blur can also occur when particles are not exactly in focus of the lenses. The maximum circle of confusion is 1.3 pixels at the edges of the observation volume.

Regarding telecenticity, we added:

The millimeter pattern calibration did not reveal any dependence on the position in the observation volume so that errors related to imperfect telecentricity of the lenses can be likely neglected.

*L 369 "D < 0.3 mm indicating that discretization errors can become substantial for D <0.3 mm": this "discretization" errors could be discussed better in the context of sizing errors.*

We added to the discussion of artificial squares:

For particles smaller than 10 px, this bias can be slightly larger due to discretization errors as can be seen from the larger impact of blur for small squares (Fig. 5.d).

**5) Error if only one camera used**

*I find it misleading to call the difference in $D_{max}$ determined from the two cameras' images a "sizing error" (L 97) or "errors in $D_{max}$ ... if only a single camera were used" (Sect 4.3 L 380-381). The difference shows how much $D_{max}$ can vary with viewing direction for a particle with a certain shape. I would argue that this is not an error. While $D_{max}$ is defined for a two-dimensional images, it seems that you assume there is a true $D_{max}$ (max of $D_{max}$ as viewed from all different directions, equivalent to $D_{max}$ if one were to define it three-dimensionally). I am not sure about a potential radar bias from $D_{max}$ that is underestimated with respect to this max $D_{max}$. Would the radar signal vary with particle orientation in a similar way as $D_{max}$ varies with orientation? Then the true radar signal could not be estimated assuming that all particles have max $D_{max}$.*

We agree with the reviewer that calling it as an error was slightly misleading, we removed that language. We rewrote the whole section to simplify the analysis and also to discuss the advantage when using tracking. However, there is indeed a "true" $D_{max}$ that is relevant for radar forward operators and when estimating mass-size relations. Particle single scattering properties in the microwave are almost always parameterized as a function of $D_{max}$ for snow. Also, it is commonly assumed that particles fall horizontally aligned so that a cloud radar really "sees" $D_{max}$. The updated section reads:

>  Here, we quantify the advantage of observing multiple orientations of a particle with the VISSS. For this, we compare one minute values of mean $D_{max}$,  $D_{eq}$, and $p$  obtained from a single camera and the observation of the leader camera alone.both observations. A positive error indicates that the observation of a single camera would be too small. For this assessmentdendritic-aggregatesanalyzed using the level1match product, which contains properties for each observed matched particle. As expected from the highly irregular shape, the errors are largest for needles. The errors peak around 0.7 mm forwith mean values of 15, -88, 18,~~

, and $AR$ change by 16%, 10%, 14%, and −12%, respectively, and when additionally considering tracking change by 24%, 19%, 24%, and −27%, respectively.  Changes for dendritic aggregates and graupel are less and surprisingly similar: $D_{max}$ increases by 8% and 7% (13% and 16%), $D_{eq}$ increases by 6% and 6% (14% and 14%), and $p$ increases by 7% and 7% (19% and 16%), respectively, when using two cameras (two cameras with tracking). The dependency of particle properties to orientation can be also seen from the fact that mean $AR$  decreases from 0.62 to 0.54 and  0.42 for aggregates and from 0.73 to 0.67 and 0.54 for graupel highlighting that orientating matters even for graupel.

Underestimating $D_{max}$ can lead to biases when using commonly used $D_{max}$ based power laws for particle mass (Mitchell, 1996) or when using in situ observations to forward model radar observations. This is because scattering properties of non-spherical particles are typically parameterized as a function of $D_{max}$ (Mishchenko et al., 1996; Hogan et al., 2012). Further, particle scattering properties are also impacted by the distribution of particle mass along the path of propagation (Hogan and Westbrook, 2014) which is impacted by $AR$. To analyze how the  different $D_{max}$ and $AR$ estimates affects the simulated radar reflectivity for vertically pointing cloud radar observations at 94 GHz, we use the  PAMTRA radar simulator (Passive and Active Microwave radiative TRAnsfer tool, Mech et al., 2020) with the riming-dependent parameterization of the particle scattering properties (Maherndl et al., 2023)  assuming horizontal particle orientation (Sassen, 1977; Hogan et al., 2002). Using two cameras (i.e., max($D_{max}$, min($AR$)) increases mean $Z_e$ values by 2.1, 2.5 and 1.8 dB for aggregates, needles, and graupel, respectively. When exploiting also the varying orientations during tracking, the offsets increase to 4.5, 4.6, and 3.7 dB, respectively, which is considerably larger than the commonly used measurement uncertainty of 1 dB for cloud radars. The change in $Z_e$ is similar to the 3.2 dB found by Wood et al. (2013) using idealized particles

*L 97-99: "Leinonen et al. (2021) found that using only a single perspective for sizing snow particles can lead to a normalized root mean square error of 6% for $D_{max}$ and Wood et al. (2013) estimated the resulting bias in simulated radar reflectivity to be 3.2 dB." Could you explain the 6% found by Leinonen 2021 (I couldn't find it by quickly looking at this reference)?*

The 6% are taken from Table 2 of Leinonen et al. (2021) by comparing $\max(D_{max})$ of the three MASC cameras to a single camera view.

*Wood 2013 refers to using disdrometer measurements of size taken instead of $D_{max}$. They estimated that $D\_disdro = 0.82\ D_{max}$, i.e. an effect of sizes 18% smaller than $D_{max}$.*

Yes, but the SVI used in the Wood et al. 2013 study uses the same size definition as the VISSS. As discussed in Appendix A of Wood et al. 2013, the offset of 3.2 dB is estimated from $D_{SVI,f} = 0.82\ D_{max}$ where $D_{SVI,f}$ is the "distance between the two furthest removed points on the SVI particle image", i.e. the maximum extent of the projected image like for a single VISSS image (see their Figure 2).

*So, I am wondering how relevant this discussion around these "sizing errors" is. Additionally, I am wondering if $D_{max}$ is always the best size to use to simulate radar signals.*

See above.

**6) Sampling volume**

*The sampling volume is well defined by the intersection of the viewing volumes of the two cameras. Thus, deriving particle concentrations should be possible. It is not clear if this is actually done (or part of future work). See also the comment about L 294-295 below: does the sampling volume depend on particle size due to the "buffer" that is removed?*

Yes, particle concentrations are retrieved as discussed in section 3.5. They are calibrated using the observation volume estimation in section 3.6. To make this more clear, we rephrased the beginning of section 3.5

> To estimate  the particle size distribution (PSD), i.e., the particle number concentration as a function of size,...

and modified section 3.6

>  Calibration of the PSD also requires to obtain the exact size of the observation volume.

**7) Clarity in descriptions in Sect. 3**

*In a few places, things remain unclear. It can be seemingly small details that make that things can be come unclear. In particular, many parts of Sect. 3 suffer from this and should be reviewed. Here are things that can be improved:*

We address the following four comments together.

*L144: specify more what ROI is here?*

*L148 "few blurred pixels around the particle that would introduce a bias": Unclear what this means.*

*L146 "commonly used background detection algorithms": What are these, algorithms to detect background?*

*L151-152: "Since filling the contour also closes potential holes in the particles, the background detection and Canny filter masks are combined": What are these two masks (only mentioned here), what is the result of this combination?*

We rewrote that paragraph to improve clarity

> Because snow may stick to the camera window, individual particles within a video frame cannot be identified by image brightness. Instead, the moving region of interest (ROI) is identified by openCV's  BackgroundSubtractorKNN class (Zivkovic and van der Heijden, 2006) in the image coordinate system (horizontal dimension $X$, vertical dimension $Y$ pointing to the ground).  The moving mask identified by the background subtraction methods cannot be used directly for particle detection because the particles in the moving foreground mask are systematically too large. For each particle, we select a 10 pixel padded box around the  region of interest (ROI) which is the smallest non-rotated rectangular box around the particle (Fig. 3). Then, we use openCV's Canny  edge detection (after applying a Gaussian blur with a standard deviation of 1.5 pixels) to identify the edges of the particle and the corresponding particle masks. To fill in small gaps in the particle contour, we  dilate the contour by 1 pixel, fill the contour, erode by 1 pixel, and identify the new contour.  This method closes potential holes in the particle mask that should be retained to avoid overestimation of particle area. Therefore, the final particle mask contains only values confirmed by the Canny filter and the background detection mask.

Note that the background detection method cited in the first draft of the paper was not the actually used one. We changed it to the correct one.

*Define AR and alpha (is AR betw 0 and 1 or >1?; alpha is angle between?)*

Added.

*L 181-182 "The minimum resolution of 1 pixel is accounted for by integrating the probability density function (PDF) for an interval of +/- 0.5 pixels."*

*What does this sentence mean? What is "minimum resolution of 1 px"?*

We expanded:

> That is, it is assumed that the difference in vertical extent $\Delta h$ (vertical position $\Delta z$) between the two cameras follows a  normally distributed probability density function (PDF) with mean zero and standard deviation 1.7 px (1.2 px), based on an analysis of manually matched particle pairs.  Since pixel measurements are discrete with 1 px steps, the PDF is integrated for an interval of $\pm$ 0.5 px.

We address the following four comments together:

*L 183 "This process requires matching the time stamps ("capture time") of both cameras": You say that matching requires "capture time", but then you match capture id instead. Then you use "recording time" to match capture id's. This is confusing.*

*L 186-190: Unclear method to find capture id offset: Why 500 frames?*

*Why "This takes advantage of the fact that only moving frames are recorded."?*

*Why max 1ms in recording time? Not using capture time, but then use time (not more than 1ms apart)?*

The time matching is indeed complicated. To address all four comments above, we expanded:

> This process requires matching the  observations of both cameras in time. The internal clocks of the cameras ("capture time")   can deviate by more than 1 frame per 10 minutes. The time assigned by the computers (" record time") is sometimes, but not always, distorted by computer load. Therefore, the continuous frame index ("capture id") is used for matching, but this requires determining the index offset between both cameras  at the start of each measurement (typically 10 minutes). For this, the algorithm uses pairs of frames with observed particles that are less than 1 ms

time  (i.e. less than 1/4 of the measurement resolution) apart in record time assuming that the lag due to computer load is only sporadically increased. This allow to identify the most common capture id offset  of the frame pairs. We found that this method gives already stable results for a subset of 500 frames.

*Can there be missing frames or varying frame rate?*

The frame rate is stable, but missing frames are possible. Therefore capture id is used instead of counting frames.

*L 194 "The joint product of the integrated PDF intervals": Is this the product of the probabilities (according to the PDFs) to have Delta h, z, i?*

Yes, we rephrased accordingly.

*Can you explain why 0.1% are falsely rejected (due to larger than normal Delta i?)?*

If the probabilities of all three parameters (h, z, i) are correct, the probability that a really observed particle has a match score below 0.001 is 0.001 (or 0.1%). This would result in a false rejection.

*In Sect 3.3 you refer to effects of misalignment but may call it "vertical alignment" or "rotation". Try to use a consistent terminology and clear and concise description.*

We now use misalignment exclusively.

*L 200-201: When is a particle observed by only one camera? Only if outside common observation volume? State that larger particles means lower ratio.*

We added "outside the common observation volume".

*"impossible" (L 206) too strong, since you then show how it can be done: Bayesian L 225 is applied to matched particles to get rotation state; matching done as described in L229-236 only using Delta h)*

Changed to

> ..., but this would  not allow to generally use the vertical position to match particles from both cameras (see above).

*Potentially confusing that you use Y_L and Z_L, and then y_L and z_L, which are not the same. Maybe mention somewhere that you use capitals for. . .*

> Note that small letters describe the three dimensional coordinate system and capital letters describe the two dimensional position on the images of

the individual camera images.

*L 215: Why can you assume psi=0? Eq. 2-4 should be simplified using psi=0.*

Thanks for pointing this out. We simplified the equation and added

> We  cannot derive $\psi$ from the observation and we have no choice but to neglect it by assuming $\psi = 0$ to reduce the number of unknowns.

*L 227: Unclear why you mention that the retrieval is overconstrained. What does it mean? What are the consequences?*

We removed the sentence.

*L229-236: The procedure is unclear, try to reformulate. Refer to Eq. (2-4) if they are used in the procedure. Is the Bayesian estimation retrieval mentioned in the previous paragraph applied in this procedure? What are "observed and retrieved particles"? How many manually selected cases? What are "all" particles?*

We shortened the paragraph to make it clearer:

> The retrieved  misalignment parameters are required for matching, but retrieving the  misalignment parameters requires matched particles. To solve this dilemma,  we use an iterative method assuming that misalignment does not change suddenly. The method starts by using the misalignment estimates and uncertainties (inflated by a factor of 10 ) from the previous time period (10 minutes) to match the particles of the current time period. These particles are used to retrieve  values for $\varphi$, $\theta$, and $O_{fz}$  which are used as a priori input for the next iteration of until the change misalignment retrieval. The iteration is stopped when the changes in $\varphi$, $\theta$, and $O_{fz}$  are less than the estimated uncertainties.

 For efficiency, the iterative method is applied only to the first 300 observed particles and the resulting coefficients are stored in the metaRotation product. A drawback of the method is that this processing step requires processing the 10-minute measurement chunks in chronological order, creating a serial bottleneck in the otherwise parallel VISSS processing chain. Obviously, this method does not work when no information is available from the previous time step, e.g., after the instrument was set up or adjusted. To get the starting point for the iteration, the matching algorithm is applied for frames where only a single, relatively large ($> 10$ px) particle is detected, so that the matching can be done based on particle height difference ($\Delta h$) alone, ignoring vertical offset ($\Delta z$).

*L 249 "pairing the particles closest in space of consecutive frames": Can this be rephrased to make it clearer?*

We rephrased

> In this example, the particle velocity is simply estimated by pairing the particles closest in space of consecutive frames.

*L 253 "The final tracking algorithm": What is algorithm used here? "pairing particles closest in space"?*

The tracking algorithm has been updated and the complete section has been rewritten

> Tracking a matched particle over time provides its three-dimensional trajectory, from which sedimentation velocity and interaction with turbulence can be determined. Since the natural tumbling of the particles provides new particle perspectives, the estimates of ~~$D_{max}$ and AR can be further improved. A proof of concept showing the potential of VISSS for velocity measurements is shown in Fig. X for a case with both needles and small rimed particles (Hyytiälä, 5 January 2022, 00:00-14:30 UTC). Needles and graupel can be distinguished using the particle complexity $c$ (Eq. 1) which is higher for needles than for graupel. In this example, the particle velocity is simply estimated by pairing the particles closest in space of consecutive frames. Still, it can be clearly seen that more complex particles (i.e. needles) fall slower than less complex particles (i.e. graupel) at the same particle size. Despite the large uncertainty of the simple velocity estimate, needle particles roughly follow a parameterization of found in for unrimed aggregates, while graupel exceeds the velocity for rimed particles in the same study. The final tracking algorithm (under development) will follow a probabilistic approach similar to particle matching. It will take into account that certain properties of a particle,particle complexity $c$, or average~~

 $A$, $p$, and $AR$ can be further improved. This can be seen in a composite of a particle (Fig.4.a-b) observed during MOSAiC, which also shows how the multiple perspectives of the particle help to identify its true shape. The example also shows that during MOSAiC the alignment of the cameras was not perfect, resulting in some of the measurements being slightly out of focus; this has been resolved for later campaigns. The tracking algorithm uses a probabilistic approach similar to particle matching taking into account that the particles' velocities only change to a certain extent from one frame to the next. That change can be quantified as a cost derived from the particles' distances and shape differences between two time steps. This allows to use the Hungarian method (Kuhn, 1955) to assign the individual matched particles to particle tracks for each time step in a way that minimizes the costs, i.e. to solve the assignment problem. To account for the fact that the particle's position is expected to change between observations, we use a Kalman filter (Kalman, 1960) to predict a particle's position based on the past trajectory and use the distance $\delta l$ between predicted and actual position for the cost estimate. Without a past trajectory, the Kalman filter uses a first guess which we derive from the velocities of previously tracked particles. We found that tracking based only on position is unstable and added the difference of particle area ($\delta A$, mean of both cameras) to the cost estimate to promote continuity of particle shape. The combined cost is estimated from the product of $\delta l$ and $\delta A$ weighted by their expected variance. The performance of the algorithm can be seen for an observation obtained in Hyytiälä on 23 January 2022 04:10 UTC where multiple particles are tracked at the same time (Fig.4.c-d). The results of the tracking algorithm are stored in the level1track product which contains the track id and the same per particle variables as the other level 1 products.

*L 259 "measurements being slightly out of focus; this has been resolved for later campaigns": Is this a result of camera alignment?*

Yes, but the location of the cameras relative to each other has been improved for later campaigns.

*L 264-265: PSDs are not binned. How is A binned with $D_{max}$ (or Deq)?*

We clarified:

> For both level2 variants, the binned PSD and $A$, perimeter $p$, and particle complexity $c$ are  available binned with $D_{max}$ and  to allow comparison with instruments using either size definition. In addition to the distributions, PSD-weighted

mean values are available for $A$, $AR$, and $c$ in addition to the first to fourth and sixth moments of the  PSD that can be used to describe normalized size distributions (Delanoë et al., 2005; Maahn et al., 2015).

*L275 "The VISSS calibration is tested . . . " could better be written as something like "The sizing capabilities of the VISSS are calibrated . . . ": How is Eq 5 used to "calibrate" $D_{max}$? The word "calibrate" seems to be the wrong word here. After the calibration, $D_{max}/um$ is determined from $D_{max}/px$ by using the equation (derived from Eq5): $D_{max}/um = (D_{max}/px$ -0.35px)\*58.75um/px.*

The sentence has been removed due to the rewritten calibration section.

*L 284-285 "Eqs. 5 and 6 are used to calibrate $D_{max}$, but only the slope is used to calibrate": Deq, perimeter, and area because potential biases from the image processing routines have not been characterized" (unclear; refer also to comments under 2) Calibration).*

The sentence has been removed due to the rewritten calibration section.

*L 288 "difference to the reference spheres is less than 2%.": What difference? Pixel resolution (slope of Eq 5)?*

Yes, this was about the slope. The calibration section has been rewritten as discussed above.

*L 289 "Part of the calibration is to. . . " doesn't seem good English.*

Changed.

*L 291 "rectangular cuboid", better use "cuboid"?*

Changed as suggested.

*L 291-292 "Therefore, the observation volumes are calculated separately for leader and follower, the eight vertices of the follower observation volume": Unclear what is done here? What are the eight vertices? Would it be better to extend the depth of the follower volume before intersection*

We extended:

>  Calibration of the PSD also requires to obtain the exact size of the observation volume. For perfectly aligned cameras, this would simply be the volume of a rectangular  cuboid with a base of 1280 px x 1280 px and a height of 1024 px. However, due to  misalignment of the cameras, the actual joint observation volume is slightly smaller than  a rectangular

cuboid and can have an irregular shape. Therefore, the observation volumes are first calculated separately for leader and follower. To calculate the intersection of the two individual observation volumes, the eight vertices of the follower observation volume are rotated to the leader coordinate system, and the OpenSCAD library is used to calculate the intersection of the two bodies is calculated using the OpenSCAD library separate observation volumes.

The eight vertices are the eight corners of the rectangular cuboid.

*L 294-295 "a buffer of $D_{max}/2$ to the edges of the image is used and the observation volume is reduced accordingly. Finally, the volume is converted from pixels to m3 using the calibration factor estimated above": What is a "buffer"? Comment on the $D_{max}/2$ buffer, i.e. particle size dependent observing volume.*

We rephrased:

> To account for the removal of partially observed particles detected at the edge of the image, a buffer of $D_{max}/2$ to the edges of the image is used and the the effective observation volume is reduced accordingly by $D_{max}/2$ px on all sides.

**Other minor things - technical corrections**

*L 111: I don't see how 600mm working distance and 250Hz results in the given pixel resolution.*

Changed.

*L 119 "rea- time" should be "real time"*

Changed.

*L 132-133 "These three processing steps comprise the level1 products": ENGLISH: object and subject swapped? Anyway, not the processing steps: level1 comprises 5 properties for each particle.*

Changed.

*L 134 "level1 observations are calibrated": What does this ("calibrated") mean?*

We added "i.e., converted from pixel in metric units"

*Fig 2.: metaRotation is missing?*

MetaRotation is in the bottom left corner.

*L 175 "XF the vertical position in the follower" seems wrong, should be "X_F is the horizontal position in the follower image"*

Changed.

*L 194 "Assuming that the probabilities for $\delta h$ and $\delta y$" (Delta y) seems wrong, should be ". . . and $\delta z$" (Delta z).*

Thanks for catching this, we reformulated the sentence.

*L204-205: The observed offsets are not constant and can change due to wind load or pressure of accumulated snow on the VISSS frame.*

*Have changes in offset and/or rotation been notice on a short time scale (due to wind load)?*

We did not observe changes on short time scales, therefore wind load is indeed unlikely. We reformulated:

> The observed offsets are not constant and can change due to  unstable surfaces or pressure of accumulated snow on the VISSS frame.

*L 213 "reader" seems wrong, should be "leader"* Changed.

*L 251-253: It cannot be clearly seen, but the cloud of points doesn't seem to follow well the shown parameterizations.* The figure has been removed.

*Inconsistently high precision: Intercepts in Eq5 and 6 (only 0.349+/-0.027) "Resolution" of VISSS2 43.13.*

Changed so that both numbers contain 5 significant digits.

*L 344 "spectra": I would be consistent in calling this PSD.*

Changed as suggested.

*L 363 "small sample size": Be more specific? Few drops? How does sample size affect DSD?*

Reformulated to:

> For larger droplets, differences are likely related to their low frequency of occurrence increasing statistical errors.

*L 399 "90deg angle to a common observation volume": Better not refer the angle to a*

*volume but: "90deg angle to each other and observe a common observation volume"*

Changed as suggested.

*L 406 "and integration of particle properties over a size distribution": Size distributions are determined in this step, NOT properties integrated over a size distribution (could be done as further step, but is not done and meant here).*

We reformulated

> The VISSS  processing steps for obtaining per-particle properties include particle detection and sizing, particle matching between the two cameras considering the exact alignment of the cameras to each other, and tracking of individual particles to estimate sedimentation velocity and improve particle property estimates. For level 2 products, the temporally averaged particle properties and size distributions are available in calibrated metric units.

*Revise also sentence with "integrated particle size distribution properties" in the Abstract.*

We simplified the abstract

> VISSS data products include  various particle properties such as  maximum extent, cross-sectional area, perimeter, complexity,  and sedimentation velocity.

---

## Author Comment (AC2)

**Introducing the Video In Situ Snowfall Sensor (VISSS)**
**Response to the reviewers**

Maximilian Maahn, Dmitri Moisseev, Isabelle Steinke,
Nina Maherndl, and Matthew D. Shupe

September 27, 2023

*Original Referee comments are in italic*

> manuscript text is indented, with added text underlined and

We would like to thank the reviewers for their very helpful comments. We revised the manuscript and responded to all of the reviewers' comments.

Besides addressing the reviewers' comments, we also included a description of the new tracking algorithm to the manuscript.

**1 Review by Charles Helms**

*This manuscript describes the VISSS instrument. VISSS is a new video-based precipitation microphysics probe designed to capture high-resolution images of snowflakes using a pair of orthogonally-pointing high-speed cameras. The manuscript also compares the VISSS measurements to those of two other precipitation microphysics probes: PIP and Parsivel. Overall, I found the manuscript to be of high quality, although there are some minor improvements that would further improve upon this quality. As I only feel minor revisions are necessary, I've opted to put all my comments (some of which are simply small typo corrections) in the order they appear in the text.*

We thank Charles Helms for the extensive review and very constructive comments.

*Line 6 (also Lines 317 and 409): Is VISSS observing up to 100,000 unique particles per minute (i.e., a falling particle is only counted once during its transit across the domain) or is it making up to 100,000 particle observations per minute (i.e., each measurement of a particle is counted, even if that particle has been measured in a previous frame)? If it's the latter, I suggest changing the wording to "100,000 particle observations per minute"*

It was indeed non-unique observations. With the new tracking algorithm, we can now say that it is up to 10.000 unique observations. We updated the wording.

*Line 6: This is the first time PIP is mentioned; suggest moving the definition of the PIP acronym from Line 8 to here.*

Changed as suggested.

*Line 52 (also Line 378): The Del Guasta (2022) reference is inserted parenthetically here instead of being in-line (similar for the Battaglia et al. reference on line 378).*

Changed as suggested.

*Line 57: The PIP acronym was already defined above, although, personally, I don't see any problem with it being defined a second time so I leave changing this up to author discretion.*

We decided to repeat definitions from the abstract.

*Line 60 (and elsewhere): Maybe a bit pedantic, but 100 microns per pixel is the pixel size rather than the resolution (i.e., the minimum resolvable particle size). Paragraph starting on Line 89: I really appreciate that the authors include the details about the camera in the text (and in table 1). The only additional piece of information the authors might consider adding is the type of camera (i.e., CCD, CMOS Global Shutter, CMOS Rolling Shutter, etc).*

We changed the wording to "pixel resolution" and added the information that it is a CMOS Global Shutter camera

*Line 113: suggest removing the word "also" to improve readability (authors' discretion)*

Changed as suggested.

*Table 1: The pixel sizes are inconsistent in their use of "." or "," as a decimal point.*

Changed.

*Table 1: If there is room for it in the table, I suggest changing "Exposure time" to "Effective exposure time" to make it clear that this is the duration of the LED being on*

*rather than the actual exposure time of the camera itself.*

Changed as suggested.

*Figure 3: It looks like the blue ellipse only appears in the final annotated image. If this is intentional, it might be worth adding a note that the fitEllipse shape is only annotated on that image otherwise it would be helpful to either note that the blue ellipse is obscured by another ellipse (and indicate which ellipse this is) or use a dashed line for the blue ellipse and put it on top of the other ellipses.*

Thanks for the suggestion, we added to the caption:

> Estimation of particle  perimeter $p$ and area $A$ (cyan), maximum dimension $D_{max}$ (via smallest enclosing circle, magenta), smallest rectangle (red), region of interest ROI (green), and elliptical fits using openCV's fitEllipseDirect (white) and fitEllipse functions (blue, covered by white line if identical to fitEllipseDirect).

*Line 159: "sphere" should be changed to "circle".*

Changed as suggested.

*Line 175: Should "vertical position" be "horizontal position"? I would think the vertical position information would be known from the leader camera. If not, this discrepancy might need a sentence or two of brief explanation.*

Thanks for catching this, changed to horizontal.

*Paragraph starting on Line 183: It's still not clear to me how the frame matching works. Is this a case of matching up the first frame that the particle appears in? If so, wouldn't this require the alignment of the two cameras to be extremely good (on the order of half a pixel or less, presumably)? Or is this more manually intensive than I'm thinking it is and the matching is based on matching up how the particle tumbles as it passes through the domain? Regardless, more details would be helpful.*

We updated the description to make it easier to follow

> This process requires matching the  observations of both cameras in time. The internal clocks of the cameras ("capture time")  can deviate by more than 1 frame per 10 minutes. The time assigned by the computers (" record time") is sometimes, but not always, distorted by computer load. Therefore, the continuous frame index ("capture id") is used for matching, but this requires determining the index offset between both cameras

 at the start of each measurement (typically 10 minutes). For this, the algorithm uses pairs of frames with observed particles that are less than 1 ms  (i.e. less than 1/4 of the measurement resolution) apart in record time assuming that the lag due to computer load is only sporadically increased. This allow to identify the most common capture id offset  of the frame pairs. We found that this method gives already stable results for a subset of 500 frames.

*(A side thought that occurred to me while reading this: have you considered matching the frames up by using a camera flash while the LEDs are obscured? I'm not sure if that'll work or not, but figured I'd mention it anyway).*

This is an interesting suggestion, but due to the telecentric principle the flash would need to replace the backlights because only light parallel to the optical axis makes it through the lens to the cameras. It could be still done by using the (also flashing) LED backlights but this would require using an external signal for cameras and flashed because the LED flashes are currently controlled by the leader camera and cannot be enabled/disabled without stopping data acquisition. However, externally triggered cameras tend to work not as reliable. But we will keep this idea in mind.

*Line 213: "reader" should be "leader"*

Changed.

*Section 3.4: I suggest removing the proof of concept tracking (and the related figure 4). It doesn't really add much to the manuscript and nearest-neighbor-based particle tracking is really bad outside of very low winds and/or very light snowfall. Depending on how much progress has been made, it might be worth adding some more information on the actual particle tracking algorithm the authors are developing.*

In the meantime, we finalized the tracking algorithm and adapted the section accordingly. The figure has been removed.

*Line 269 – 271: If multiple observations of a single particle are included in the PSD, wouldn't this bias the PSD towards slower falling particles?*

For the PSD, we are interested how many particles are *on average* in a our observation volume. Because smaller particles remain longer in the observation volume, we do not have a bias in our observation.

*Line 313: It took me several times reading these sentences to realize the "Fig 6. c" did not refer to panel c of figure 6, but that these are two completely separate thoughts; suggest replacing "c" with "Particle complexity" or "Particle complexity c" to avoid this*

*issue.*

We rephrased the sentence.

*Line 314: insert "particles" after "weighted to smaller".*

Added

*Lines 336 – 354 (regarding the PIP underestimation of N0\*): I don't think the dilation is the issue here. PIP's processing applies an edge detection filter, dilates the resulting image twice (using the 3x3 kernel), fills in any holes, and then erodes the hole-filled image twice to (theoretically) undo the dilation step (using the same 3x3 kernel). A few possibilities that come to my mind to explain the discrepancies between VISSS and PIP are: 1) the dilation is merging nearby needles into a single particle, thus decreasing the number of small particles (presumably this would be paired with an increase in larger particles); 2) the image compression (which averages vertically-adjacent pixels to reduce the data rates) is essentially destroying the smaller needles; and 3) that the dilation and hole-filling of higher complexity particles is artificially inflating the equivalent diameter and this is introducing a bias into the PSD moments used to compute N0\* (presumably this one is less relevant for the needles, but later you mention PIP having issues with high complexity particles, so I included this as well). Unfortunately, I don't have any deeper insight as to which one of these might be the culprit (if it even is one of these), but I'll add this to the top of my list of PIP behaviors to look into. Either way, I don't think it would be the dilation itself as that adds to the particles.*

Thanks for identifying this mistake, it looks like we mixed up dilation and erosion. Since analyzing the PIP image processing is outside the scope of this paper and a discussion would be highly speculative without additional insight into the PIP processing, we removed the section.

*From the VISSS side of things, if the PSD is being biased towards slower falling particles by including all particle observations, this might produce a bias at small particles in the VISSS PSD. It should be relatively easy to test this by comparing the VISSS PSD as it appears here to the PSD that would be produced if only particles appearing in every 25th frame are included in the PSD (this is how PIP computes its PSD, for reference). Under normal conditions, 25 frames should be more than enough time for any particles observed in a frame to exit the measurement volume.*

To our knowledge, the PIP uses every frame for estimating the PSD, but this might be related to different software versions. But we are not sure whether we understand the reviewer correctly because it is unclear to us how using every 25th frame should change the PSD except by adding noise. If our goal was to measure how many particles fall through a given area per time interval, the reviewer would be right and we would need to remove all but one observation of a particle observed multiple times. However, we want to observe how many particles there are *on average* in our observation volume which

requires counting particles multiple times as long as they are in the observation volume and dividing by the number of frames in the end.

*Line 337: should "width" be "length"? In my mind the width of a particle is more closely aligned with (one of) the shorter axis of a particle, but when reading this it feels like the authors are referring to the longer axis of the needles.*

No, we are referring to the shorter axis.

*Lines 342 – 343 and 352 – 353: As I mentioned above, dilation shouldn't result in the removal of any parts of a snowflake as dilation expands the particles. Erosion could, in theory, remove parts, but I doubt it since the erosion step occurs after the particles have already been dilated twice.*

As discussed before, the text has been removed.

*Fig. 6: It might be helpful to also include the observed particles per minute for the PIP and Parsivel just to give a point of comparison. For PIP at least, the PSD is computed using the \*_a_p_60.dat files (which only includes particles observed during every 25th frame to avoid double counting).*

Good idea, we changed the figure as suggested.

*Additionally, if the authors' wish to, the particle complexity can be computed from the PIP files by dividing the particle area by the hydraulic radius (Hy_Rad), which is the ratio between area and perimeter to get the particle perimeter and then plugging the relevant values into Eq 1. That said, as discussed in Helms et al. (2022) [section 3.1], the PIP software takes some potentially questionable (when applied to a snowflake) shortcuts when computing the perimeter, so comparing particle complexity between VISSS and PIP may not be particularly informative as to the accuracy of VISSS. I am less familiar with the specifics of the Parsivel output files, but similar methods may be possible there as well. Either way, I leave choice of adding this up to the authors' discretion.*

Given the background on the PIP data processing the reviewer provided, we decided not comparing complexity observations of PIP and VISSS.

*Fig. 7: It's hard to make anything out on the image due to the small size of each frame. If might be beneficial to include a subset of these to make each frame larger so readers can better appreciate the resolution of the cameras.*

We updated the figure as suggested.

*Fig. 8: It would be helpful to have the time period over which the PSDs are computed for each of the instruments either in the caption and/or the text (apologies if this is in the text and I missed it, I added this comment after having read through the paper and didn't see it mentioned when looking back through again).*

We added the information that the distributions were integrated over one minute to the caption.

*Line 432: I certainly appreciate VISSS being open source! We (collectively) shouldn't have to pull teeth to understand how instruments produce their measurements.*

This is also our motivation, we thank the reviewer for this comment.

*Appendix A: It took a couple attempts, but I was able to replicate the derivation in Appendix A.*

We appreciate the effort of double checking the derivation.

---

## Referee Report (RR1)

Review of Introducing the Video In Situ Snowfall Sensor (VISSS)
Reviewed by Charles Helms

The authors have satisfactorily addressed all of the comments I raised in my previous review and I feel like this manuscript is ready for publication.  That said, there are a few very minor suggested corrections (almost exclusively typos) that I have included at the end of this review.

Following up to the authors' response to my comment on the PSD and slower falling particles: Thank you for the explanation regarding the interpretation of the PSD not as the number of particles that fall through the volume in a given time but as the average number of particles in a volume at any given time. My work with the PIP and other instruments has mostly focused on the measurements themselves rather than the resulting PSD.  Not sure about the PIP subsampling, but it's not particularly relevant to the manuscript anyway.  That said, it does seem odd that the PIP data would be subsample given the goal is the average number of particles in the volume at any given time; perhaps there was some other reason I'm not aware of (or perhaps the person who told me that the PIP PSD used the subsampled data was mistaken).

**Minor Comments:**

Line 198: "This allow to identify…" should be something like "This allows us to identify…" or "This allows the algorithm to identify…"

Line 199: "We found that this method gives already stable results…"; suggest removing "already"

Line 205: "campaign" should be plural

Line 218: "…but this would not allow to generally…"; add "us" (or similar word) after "allow"

Line 262-263: The authors might also want to mention how the tracking is initialized when there are no previously tracked particles (e.g., at start up).  I assume this is taken care of either alongside the camera alignment steps or via some default value, but it might be a good idea to state things explicitly.

Line 308: "…but the offset is this time negative…"; suggest removing "this time"

Line 423: "…probably related to problems of the PIP image processing."; the wording is a bit awkward, suggest something like "probably as a result of the PIP image processing implementation" or "probably as a result of limitations in the PIP image processing implementation"

Line 449: "When exploiting also the varying orientations during tracking" sounds awkward.  Suggest something like "When the varying orientations are taken into account"

---

## Referee Report (RR2)

I thank the authors for carefully considering all the feedback. I appreciate that they improved many descriptions and some algorithms resulting in a clearer manuscript.

Thank you for demonstrating the optical resolution. I can see almost all lines of the microscope slide you are showing. So, it is fair to say the resolution is on the order of 50µm given that these lines have a similar thickness.

Regarding small particles, I agree with your concern that you are likely "loosing" (not detecting) some of the smallest particles during image processing. I am still curious, however, to see examples of 2-px particles alongside their contour and Dmax, A, p values. As you have published data, I will have a look and do not suggest adding anything to the paper.

In the following I am only asking for a few clarifications, which mostly refer to changed sections. I am referring to line numbers of manuscript-version2.

**Sect 3.1 Particle Detection**

L153: "… particles in the moving foreground mask are systematically too large."
You talk about a moving ROI, then moving mask, and eventually about moving foreground mask. I am not sure what is the "particle" at this stage? Or do you mean the ROI is larger than the particle?
L154: Replace "region of interest (ROI)" with "ROI".
L154: "non-rotated". Is there a better way to define this type of smallest rectangular box? (major axis along x or y?)
L156-159: What is a "gap in the contour"? For me a contour is a continuous line. It is hard to follow exactly what is done here. Perhaps and illustrated example of what could happen and how it is prevented would help a lot (could be in Appendix B).

**Sect 3.2 Particle Matching,** L 191-192: The sentence "Since pixel measurements are discrete with 1 px steps, the PDF is integrated for an interval of ±0.5 px" for me omits why the PDF is integrated. This may be obvious for some, but for clarity I would anyhow include it (correct me if I am wrong):
"To determine the probability (of, for example, a certain vertical extent), the PDF is integrated over an interval of ±0.5 px (representing the discrete 1-px steps)."

**Sect 3.4 Particle Tracking**, L358-359 "shape difference": Shape refers to area here? Better say "area difference" then.

**Sect 3.4**, Fig 4c,d. It took me some time to understand what exactly is shown. Perhaps small changes in the caption can improve it:
"… shows a frame of the leader (c) and the matched frame of the follower (d). …
For each particle (surrounded by boxes) the particle track is shown. The tracks indicate past …"

**Sect 3.5**, L270-273: This sentence suggests that the PSD is a property averaged in size bins. Isn't it instead the number concentration in size bins (normalized with the bin width)?
So, I would suggest being correct and clearer by saying something like (guessing how you determine concentration and account for size dependent observation volume, see comment on L335-336 below):
"To estimate the particle size distribution (PSD), i.e., the particle number concentration as a function of size, the individual particle data are binned by particle size (1 px spacing, i.e. 43.125 or 58.75 .m) and the number of particles in the bins are divided by the observation volume. These binned number concentrations are then averaged over all frames during one-minute periods. Then also binned particle properties such as area and perimeter are averaged to one minute resolution for."
Correct me if I am wrong and try to improve sentence accordingly.

**Sect 3.6 Calibration**, L330,331: From the Response Comments you seemed to agree that it was sufficient to say "cuboid". But I see twice the term "rectangular cuboid".

**Sect 3.6 Calibration**, L 336-336: " To account for the removal of partially observed particles detected at the edge of the image, the effective observation volume is reduced by Dmax/2 px on all sides."
This means that the observation volume is size dependent. What if two or more particles are in the observation volume, how is concentration calculated (as I guessed above, see comment on L270-273)? It may be good to mention the size dependence and how you take care about it (for example referring to Sect 3.5) or how it affects results, if it does). With that, you also make it clear that and how you use the observation volume, for which you just described how to determine it.

**Sect 4.1**
I would extend the sentence ending in L384 for clarity (of what 50% advantage means):
"... reduced to 50% more particles than observed by Parsivel and PIP."

**Sect 4.3**, L439 "orientating": Would "orientation" be better?

**Technical**:
I would, according to standards, use roman font (not italics) for indices that are descriptive (i.e. do not refer to other variables): $D_{max}$, $D_{eq}$, $X_L$, ...
Check for inconsistent use of font (variables that appear in both italics and roman): N0*, $D_{max}$ and $D_{eq}$ (Fig8 caption), $D_{32}$ (is $D_{23}$ in Fig6 caption).

---

## Author Response (AR2)

**Introducing the Video In Situ Snowfall Sensor (VISSS)**
**Response to the reviewers**

Maximilian Maahn, Dmitri Moisseev, Isabelle Steinke,
Nina Maherndl, and Matthew D. Shupe

November 21, 2023

*Original Referee comments are in italic*

> manuscript text is indented, with added text underlined and

*Original Referee comments are in italic*

We thank the reviewers and the editor for their—again—very helpful comments. We responded to all comments and changed the manuscript accordingly.

**1  Review by Thomas Kuhn**

*I thank the authors for carefully considering all the feedback. I appreciate that they improved many descriptions and some algorithms resulting in a clearer manuscript. Thank you for demonstrating the optical resolution. I can see almost all lines of the microscope slide you are showing. So, it is fair to say the resolution is on the order of 50µm given that these lines have a similar thickness. Regarding small particles, I agree with your concern that you are likely "loosing" (not detecting) some of the smallest particles during image processing. I am still curious, however, to see examples of 2-px particles alongside their contour and Dmax, A, p values. As you have published data, I will have a look and do not suggest adding anything to the paper. In the following I am only asking for a few clarifications, which mostly refer to changed sections. I am referring to line numbers of manuscript-version 2.*

We thank Thomas Kuhn for taking the time to review the paper again and his helpful comments.

*Sect 3.1 Particle Detection*
*L153: "... particles in the moving foreground mask are systematically too large." You talk about a moving ROI, then moving mask, and eventually about moving foreground mask. I am not sure what is the "particle" at this stage? Or do you mean the ROI is larger than the particle?*
*L156-159: What is a "gap in the contour"? For me a contour is a continuous line. It is hard to follow exactly what is done here. Perhaps and illustrated example of what could happen and how it is prevented would help a lot (could be in Appendix B).*

Both comments are addressed together: We reworded the paragraph to use the word contour more precisely and to name the moving mask more consistently

> Instead, the moving  mask of pixels is identified by openCV's BackgroundSubtractorKNN class (Zivkovic and van der Heijden, 2006) in the image coordinate system (horizontal dimension $X$, vertical dimension $Y$ pointing to the ground).  In the moving mask identified by the background subtraction  method, the individual particles are systematically too large so that the moving mask cannot be used directly for particle sizing. For each particle, i.e. connected group of moving pixels, we select a 10  px padded box around the region of interest (ROI) which is the smallest non-rotated rectangular box around the particle's moving mask (Fig. 3).  This extended ROI is the input for openCV's Canny edge detection (after applying a Gaussian blur with a standard deviation of 1.5 px) to identify the edges of the particle. To estimate the particle mask by filling in the retrieved particle edges, gaps (typically 1 px in size) between the particle edges must be closed. For this, we dilate the  retrieved edges by 1  px to form a closed contour, fill in the created contour, and erode the filled shape by 1  px to obtain the particle mask. To detect potential particle holes, which should be retained to avoid  overestimating the particle area, the  Canny filter particle mask and the moving mask are combined for the final particle mask.

*L154: "non-rotated". Is there a better way to define this type of smallest rectangular box? (major axis along x or y?)*

Yes, this is actually estimated by the minAreaRect algorithm which is one of the algorithms used for determining the aspectRatio. But the ROI is more a technical step required for determining the region where other algorithms like blur estimation, Canny edge detection etc. are applied to.

*L154: Replace "region of interest (ROI)" with "ROI".*

Not changed, because the previous mention of ROI was removed.

*Sect 3.2 Particle Matching, L 191-192: The sentence "Since pixel measurements are discrete with 1 px steps, the PDF is integrated for an interval of ±0.5 px" for me omits why the PDF is integrated. This may be obvious for some, but for clarity I would anyhow include it (correct me if I am wrong): "To determine the probability (of, for example, a certain vertical extent), the PDF is integrated over an interval of ±0.5 px (representing the discrete 1-px steps)."*

Changed to

>  To determine the probability (of e.g., measuring a certain vertical extent), the PDF is integrated  over an interval of  ±0.5 px representing the discrete 1 px steps.

*Sect 3.4 Particle Tracking, L358-359 "shape difference": Shape refers to area here? Better say "area difference" then.*

Changed as suggested.

*Sect 3.4, Fig 4c,d. It took me some time to understand what exactly is shown. Perhaps small changes in the caption can improve it: "... shows a frame of the leader (c) and the matched frame of the follower (d). ... For each particle (surrounded by boxes) the particle track is shown. The tracks indicate past ..."*

Changed as suggested.

*Sect 3.5, L270-273: This sentence suggests that the PSD is a property averaged in size bins. Isn't it instead the number concentration in size bins (normalized with the bin width)? So, I would suggest being correct and clearer by saying something like (guessing how you determine concentration and account for size dependent observation volume, see comment on L335-336 below): "To estimate the particle size distribution (PSD), i.e., the particle number concentration as a function of size, the individual particle data are binned by particle size (1 px spacing, i.e. 43.125 or 58.75 .m) and the number of particles in the bins are divided by the observation volume. These binned number concentrations are then averaged over all frames during one-minute periods. Then also binned particle properties such as area and perimeter are averaged to one minute resolution for." Correct me if I am wrong and try to improve sentence accordingly.*

We agree that this was formulated in a confusing way. To improve readability, we renamed the section from Particle size distributions to Time-resolved properties and moved it behind the calibration section. Also, we moved the estimation of the observation volume to the description of the PSD so that both are discussed together:

>  Time-resolved particle properties
>
> While Level 1 products contain per-particle properties, Level 2 products provide time-resolved properties. This includes the particle size distribution (PSD), which is the concentration of particles as a function of size normalized to the bin width. To estimate the PSD, the individual particle data are binned by particle size (1 px spacing, i.e. 43.125 or 58.75 µm), averaged over all frames during one-minute periods, and divided by the observation volume. For perfectly aligned cameras,  the observation volume would simply be the volume of a  cuboid with a base of 1280 px x 1280 px and a height of 1024 px. However, due to misalignment of the cameras, the actual joint observation volume is slightly smaller than a  cuboid and can have an irregular shape. Therefore, the observation volumes are first calculated separately for leader and follower. To calculate the intersection of the two individual observation volumes, the eight vertices of the follower observation volume are rotated to the leader coordinate system, and the OpenSCAD library is used to calculate the intersection of the two separate observation volumes in pixel units. To account for the removal of partially observed particles detected at the edge of the image, the effective observation volume is reduced by  $D_{\max}/2$ px on all sides. Consequently, each size bin of the PSD is calibrated independently with a different, $D_{\max}$-dependent effective observation volume. Finally, the volume is converted from pixel units to $m^3$ using the calibration factor estimated above.

The rest if the paragraph has been mostly flagged as new by latexdiff because it was moved from section 3.5

> The Level 2 products are available based on the level1match and level1track products. For level2match, binned particle properties are available either from one of the cameras or using the minimum, average or maximum from both cameras for each observed particle property. This means that the multiple observations of the same particle all contribute to the PSD. This does not bias the PSD because the number of observed particles is divided by the number of frames, and the PSD describes how many particles are *on average* in the observation volume. For level2track, the distributions are based on the observed tracks instead of individual particles, and are calculated using the minimum, maximum, mean, or standard deviation along the observed track using both cameras. The use of the maximum (minimum) value along a track is motivated by the assumption that the estimated

properties of a particle such as $D_{\max}$ (AR) of a particle will be closer to the true value than when ignoring the different perspectives of a particle along the track obtained by the two cameras.

For both level2 variants, the binned PSD and $A$, perimeter $p$, and particle complexity $c$ are available binned with $D_{\max}$ and $D_{\mathrm{eq}}$ to allow comparison with instruments using either size definition. In addition to the distributions, PSD-weighted mean values with one minute resolution are available for $A$, $AR$, and $c$ in addition to the first to fourth and sixth moments of the PSD that can be used to describe normalized size distributions (Delanoë et al., 2005; Maahn et al., 2015).

*Sect 3.6 Calibration, L330,331: From the Response Comments you seemed to agree that it was sufficient to say "cuboid". But I see twice the term "rectangular cuboid".*

We must have overlooked that, changed as suggested.

*Sect 3.6 Calibration, L 336-336: " To account for the removal of partially observed particles detected at the edge of the image, the effective observation volume is reduced by Dmax/2 px on all sides." This means that the observation volume is size dependent. What if two or more particles are in the observation volume, how is concentration calculated (as I guessed above, see comment on L270-273)? It may be good to mention the size dependence and how you take care about it (for example referring to Sect 3.5) or how it affects results, if it does). With that, you also make it clear that and how you use the observation volume, for which you just described how to determine it.*

We moved that paragraph to section 3.6 and added the following sentence:

Consequently, each size bin of the PSD is calibrated independently with a different, $D_{\max}$-dependent effective observation volume.

*Sect 4.1 I would extend the sentence ending in L384 for clarity (of what 50% advantage means): "... reduced to 50% more particles than observed by Parsivel and PIP."*

Changed as suggested.

*Sect 4.3, L439 "orientating": Would "orientation" be better? Technical: I would, according to standards, use roman font (not italics) for indices that are descriptive (i.e. do not refer to other variables): Dmax, Deq, XL, ... Check for inconsistent use of font (variables that appear in both italics and roman): N0*, Dmax and Deq (Fig8 caption), D32 (is D23 in Fig6 caption).*

Changed as suggested.

**2 Review by Charles Helms**

*The authors have satisfactorily addressed all of the comments I raised in my previous review and I feel like this manuscript is ready for publication. That said, there are a few very minor suggested corrections (almost exclusively typos) that I have included at the end of this review.*

*Following up to the authors' response to my comment on the PSD and slower falling particles: Thank you for the explanation regarding the interpretation of the PSD not as the number of particles that fall through the volume in a given time but as the average number of particles in a volume at any given time. My work with the PIP and other instruments has mostly focused on the measurements themselves rather than the resulting PSD. Not sure about the PIP subsampling, but it's not particularly relevant to the manuscript anyway. That said, it does seem odd that the PIP data would be subsample given the goal is the average number of particles in the volume at any given time; perhaps there was some other reason I'm not aware of (or perhaps the person who told me that the PIP PSD used the subsampled data was mistaken).*

We thank Charles Helms for taking the time to review the paper again and his helpful comments.

*Line 198: "This allow to identify..." should be something like "This allows us to identify..." or "This allows the algorithm to identify..."*
*Line 199: "We found that this method gives already stable results..."; suggest removing "already"*
*Line 205: "campaign" should be plural*
*Line 218: "...but this would not allow to generally..."; add "us" (or similar word) after "allow"*

All changed as suggested.

*Line 262-263: The authors might also want to mention how the tracking is initialized when there are no previously tracked particles (e.g., at start up). I assume this is taken care of either alongside the camera alignment steps or via some default value, but it might be a good idea to state things explicitly.*

We added

> Without a past trajectory, the Kalman filter uses a first guess which we derive from the velocities of 200 previously tracked particles. If no previous particles are available, the tracking algorithm is applied twice to the first 400 particles to avoid a potential bias caused by using a not case-specific fixed value as a first guess.

*Line 308: "...but the offset is this time negative..."; suggest removing "this time"*

*Line 423: "...probably related to problems of the PIP image processing."; the wording is a bit awkward, suggest something like "probably as a result of the PIP image processing implementation" or "probably as a result of limitations in the PIP image processing implementation"*

*Line 449: "When exploiting also the varying orientations during tracking" sounds awkward. Suggest something like "When the varying orientations are taken into account*

All changed as suggested.